# Polyphenols, Alkaloids, and Terpenoids Against Neurodegeneration: Evaluating the Neuroprotective Effects of Phytocompounds Through a Comprehensive Review of the Current Evidence

**DOI:** 10.3390/metabo15020124

**Published:** 2025-02-13

**Authors:** Enzo Pereira de Lima, Lucas Fornari Laurindo, Vitor Cavallari Strozze Catharin, Rosa Direito, Masaru Tanaka, Iris Jasmin Santos German, Caroline Barbalho Lamas, Elen Landgraf Guiguer, Adriano Cressoni Araújo, Adriana Maria Ragassi Fiorini, Sandra Maria Barbalho

**Affiliations:** 1Department of Biochemistry and Pharmacology, School of Medicine, Universidade de Marília (UNIMAR), Marília 17525-902, São Paulo, Brazil; 2Laboratory of Systems Integration Pharmacology, Clinical and Regulatory Science, Research Institute for Medicines, Universidade de Lisboa (iMed.ULisboa), Av. Prof. Gama Pinto, 1649-003 Lisbon, Portugal; 3HUN-REN-SZTE Neuroscience Research Group, Danube Neuroscience Research Laboratory, Hungarian Research Network, University of Szeged (HUN-REN-SZTE), Tisza Lajos Krt. 113, H-6725 Szeged, Hungary; 4Department of Biological Sciences (Anatomy), School of Dentistry of Bauru, University of São Paulo (FOB-USP), Alameda Doutor Octávio Pinheiro Brisolla, 9-75, Bauru 17012-901, São Paulo, Brazil; 5Department of Gerontology, School of Gerontology, Universidade Federal de São Carlos (UFSCar), São Carlos 13565-905, São Paulo, Brazil; 6Postgraduate Program in Structural and Functional Interactions in Rehabilitation, University of Marília (UNIMAR), Marília 17525-902, São Paulo, Brazil; 7Department of Biochemistry and Nutrition, School of Food and Technology of Marília (FATEC), Marília 17500-000, São Paulo, Brazil

**Keywords:** neuroprotection, Alzheimer’s disease, Parkinson’s disease, neurodegeneration, phytochemicals, neuroinflammation, polyphenols, antioxidant, anti-inflammatory

## Abstract

Neurodegenerative diseases comprise a group of chronic, usually age-related, disorders characterized by progressive neuronal loss, deformation of neuronal structure, or loss of neuronal function, leading to a substantially reduced quality of life. They remain a significant focus of scientific and clinical interest due to their increasing medical and social importance. Most neurodegenerative diseases present intracellular protein aggregation or their extracellular deposition (plaques), such as α-synuclein in Parkinson’s disease and amyloid beta (Aβ)/tau aggregates in Alzheimer’s. Conventional treatments for neurodegenerative conditions incur high costs and are related to the development of several adverse effects. In addition, many patients are irresponsive to them. For these reasons, there is a growing tendency to find new therapeutic approaches to help patients. This review intends to investigate some phytocompounds’ effects on neurodegenerative diseases. These conditions are generally related to increased oxidative stress and inflammation, so phytocompounds can help prevent or treat neurodegenerative diseases. To achieve our aim to provide a critical assessment of the current literature about phytochemicals targeting neurodegeneration, we reviewed reputable databases, including PubMed, EMBASE, and COCHRANE, seeking clinical trials that utilized phytochemicals against neurodegenerative conditions. A few clinical trials investigated the effects of phytocompounds in humans, and after screening, 13 clinical trials were ultimately included following PRISMA guidelines. These compounds include polyphenols (flavonoids such as luteolin and quercetin, phenolic acids such as rosmarinic acid, ferulic acid, and caffeic acid, and other polyphenols like resveratrol), alkaloids (such as berberine, huperzine A, and caffeine), and terpenoids (such as ginkgolides and limonene). The gathered evidence underscores that quercetin, caffeine, ginkgolides, and other phytochemicals are primarily anti-inflammatory, antioxidant, and neuroprotective, counteracting neuroinflammation, neuronal oxidation, and synaptic dysfunctions, which are crucial aspects of neurodegenerative disease intervention in various included conditions, such as Alzheimer’s and other dementias, depression, and neuropsychiatric disorders. In summary, they show that the use of these compounds is related to significant improvements in cognition, memory, disinhibition, irritability/lability, aberrant behavior, hallucinations, and mood disorders.

## 1. Introduction

Neurodegenerative diseases are a heterogeneous group of chronic, usually age-related disorders characterized by progressive neuronal loss, deformation of neuronal structure, or loss of neuronal function. They remain a significant focus of scientific and clinical interest due to their increasing medical and social importance. Most neurodegenerative diseases present intracellular protein aggregation or their extracellular deposition (plaques), such as α-synuclein (α-Syn) in Parkinson’s disease (PD), amyloid beta (Aβ)/tau aggregates in Alzheimer’s disease (AD), superoxide dismutase (SOD) in amyotrophic lateral sclerosis (ALS), or huntingtin protein with glutamine repeats in Huntington’s disease (HD) [1,2]. Another issue is that one of the facets of neurodegeneration is the malfunction of specific cells. In this sense, subarachnoid hemorrhage (SAH), characterized by the rupture of intracranial blood vessels causing the direct influx of blood into the subarachnoid space, is linked, for example, to microglia dysfunction. Although this cell develops diverse functions such as maintaining the neural environmental balance, supporting neurons, mediating apoptosis, involvement in immune regulation, and exerting neuroprotective effects, it also has a fundamental role in the pathogenesis of this vascular alteration since its exacerbated activity can weaken vessels and cause inflammatory changes, facilitating the occurrence of SAH [3,4].

Neurodegeneration affects millions of people around the world, causing psychological and socioeconomic burns [5,6]. This condition refers explicitly to symptoms encompassing memory, reasoning, and social skills. As an example, AD, vascular dementia (VD), dementia with Lewy bodies (DLB), frontotemporal dementia (FTD), and mixed dementia (MD) are characterized as progressive. At the same time, PD, HD, and traumatic brain injury (TBI) are related to dementia [7,8]. These changes are products of impaired mitochondrial signaling, damaged epigenetic modulations, genomic instability, cellular senescence, altered cell-to-cell communication, telomere shortening, and dysregulated nutrient sensing, all of which promote inflammation, oxidative stress, and permanent nerve damage [9,10,11,12]. It is worth mentioning that certain factors help in the onset of neurodegeneration, such as age and family history, alcohol and smoking, nutrition, physical activity and lifestyle, sleep disorders, diabetes, cardiovascular health, and environmental risk factors, causing characteristics such as memory loss, disorientation, confusion and difficulty in communication, coordination, organization, and motor functions, psychological changes, including depression, anxiety, paranoia, inappropriate behavior, and the manifestation of personality changes [13,14].

An excess of reactive oxygen species (ROS) and reactive nitrogen species, which promote oxidative stress, results in cellular injury, i.e., cellular disorders, including neurodegenerative diseases such as AD [15]. In addition to tissue damage, oxidative stress can also influence blood–brain integrity, promoting neuroinflammation, which is an early stage in the pathophysiology of AD [16]. It is worth highlighting that oxidative stress, through ROS, can cause dopaminergic neuronal damage and loss in PD. Thus, the dysregulation of cellular redox activity leads to a higher release of ROS than the endogenous antioxidant activity can metabolize. After this, neuronal damage begins, and oxidative damage occurs to deoxyribonucleic acid (DNA), proteins, or lipids, with oxidants and superoxide radicals produced in the mitochondria [17,18,19]. Furthermore, inflammation-induced neurodegeneration can promote the progression of central nervous system diseases such as those mentioned above. The inflammatory process causes autoreactive T cells to infiltrate the central nervous system, which can cause demyelination, neuroaxonal damage, protein loss, electrolyte imbalance, and other processes, causing the progressive loss of brain volume, neurological deficits, and even loss of the spinal cord [20]. Finally, problems in mitochondrial DNA and the respiratory chain are linked to the pathogenesis of many neurodegenerative disorders [21]. This occurs because the central nervous system is susceptible to mitochondrial dysfunction, as the brain is highly metabolically active and, therefore, susceptible to bioenergetic failure. Secondly, the brain has fewer antioxidant defenses against oxidative stress, and neurons are post-mitotic and irreplaceable, meaning any neuronal injury will be fatal to the cell. Therefore, nerve cells have a consistently high demand for adenosine triphosphate (ATP) produced via mitochondrial metabolism. However, although mitochondria are essential for generating ATP when deregulated, they can produce iron–sulfur clusters and calcium, cause cell death, and amplify the signaling of ROS, potentiating neurogenerative diseases [22,23].

These conditions are generally related to increased oxidative stress and inflammation, so phytocompounds can help prevent or treat neurodegenerative diseases. These compounds can include flavonoids (luteolin and quercetin), alkaloids (berberine, huperzine A, and caffeine), terpenoids, phenolic acids (caffeic acid and ferulic acid), and other general polyphenols (resveratrol and pterostilbene).

The use of phytocompounds in the treatment of neurodegenerative diseases consists of using antioxidant and inhibitory properties on certain neurotransmitter-degrading enzymes, causing an increase in the concentration and activity of other types of neurotransmitters, reducing neuroinflammation, protecting defense cells such as microglia, and causing its activity to be enhanced to a certain extent so as not to overload it, delay the progression of dementia, improve pro- and anti-apoptotic properties, at the same time that it can reduce possible side effects, due to the decreasing concentrations used in treatments [8,24,25,26]. Previous reports have diligently reviewed the impact of bioactive compounds and dietary phytochemicals against neurodegeneration. To date, Venktesan et al. [27] examined the roles of (−)-3,5-dicaffeoylmucoquinic acid, quinic acid, ligraminol, juniperigiside, and other compounds against neurodegeneration. However, their analysis delved into preclinical research. It did not focus on clinical trials, limiting the generalizability of their findings and not familiarizing the scientific community with translational research. Additionally, they solely concentrated on the influence of phytochemicals on neurotrophins and did not evaluate the potential role of these bioactive constituents against neuroinflammation and other critical aspects of neurodegeneration. On the other hand, Sahebnasagh et al. [28] examined the effects of neurohormetic phytochemicals in the occurrence of neurodegenerative diseases. However, their analysis focused mainly on the neurohormetic dose–response concepts of phytochemicals against neurodegeneration, lacking sufficient discussion on the underlying neuroprotection mechanisms of bioactive compounds against neuroinflammation, protein aggregation, and synaptic dysfunctions, critical aspects of neurodegeneration intervention with phytochemicals. Due to the increase in neurodegenerative diseases, this review aims to show the role of polyphenols, alkaloids, and terpenoids in these conditions. To the best of our knowledge, this is the first comprehensive review to fully elucidate the potential neuroprotective benefits of these classes of phytocompounds against neurodegeneration based mainly on clinical trials. An excellent body of research on the neuroprotective effects of polyphenols, alkaloids, and terpenoids provides a strong foundation for their investigation in translational research within clinical trials. However, no previous review has addressed this potential. Our review aims to fill this gap in the literature, paving the way for future research directions and the complete translation of phytotherapy into daily clinical practice.

## 2. Neurodegenerative Conditions

AD shows a gradual decline in memory, thinking, behavior, and social skills progressively and irreversibly. It is one of the leading causes of dementia among neurodegenerative diseases, profoundly impacting the physical well-being of the patient. Multiple etiological factors may trigger the disease, including age, genetics, environmental factors, lifestyle, and health conditions, which can act individually or synergistically [29,30]. Its pathogenesis is primarily linked to excessive oxidative stress due to mitochondrial dysfunction or an imbalance between the ROS formation system and cellular antioxidant agents, including catalase (CAT), SOD, and glutathione peroxidase (GPx). It is also associated with the accumulation of Aβ plaques, disturbances in calcium homeostasis, the accumulation of metal ions, and the intracellular generation of neurofibrillary tangles as a result of excessive phosphorylation of tau protein. Vascular factors and even erroneous immune responses, such as pro-inflammatory activation of microglia that increases the conversion of resting astrocytes into reactive astrocytes, also play a role. All these associated factors lead to a cascade that intensifies inflammation, neurodegeneration, and disruption of neuronal signaling pathways [31,32,33]. Furthermore, the liver can exhibit a central role in the metabolism of Aβ, as bile acids (BAs) assist in breaking down circulating Aβ, facilitating its uptake by hepatocytes. In other words, the liver cleans Aβ directly through hepatocyte-mediated degradation or indirectly by modulating the levels of plasma transport proteins and cholesterol metabolism related to Aβ. Therefore, negative regulation of this clearance mechanism in the liver, whether due to cirrhosis or hepatitis, can lead to or intensify the accumulation of Aβ in the brain and, consequently, the progression of AD. Epidemiologically, 35 million people had AD in 2010. However, the number of dementia patients can double every 20 years, reaching 65.7 million in 2030 and 115.4 million in 2050 [34,35,36].

PD is a progressive and irreversible movement-related neurodegenerative disorder that mainly affects the nigrostriatal system and is characterized by resting rigidity, tremors, bradykinesia, and ataxia [37]. In addition, it may present at the same time with non-motor symptoms, such as depression, constipation, and olfactory disorders. Its etiology may be linked to environmental toxins, aging, and even genetic factors, of which some genes are more important, such as α-Syn gene (SNCA), Leucine-rich repeat kinase 2 (LRRK2), PTEN-induced kinase 1 (PINK1), parkin RBR E3 ubiquitin-protein ligase (PARKIN), acid beta-glucosidase (GBA), ATPase cation transport 13A2 (ATP13A2), VPS35 retromer complex component (VPS35), F-box protein 7 (FBXO7), and Protein deglycase DJ-1 (DJ-1), which cause changes in autophagic/lysosomal pathways, impairing the complete control of intracellular degradation of misfolded proteins [38,39]. Pathologically, PD presents the formation of Lewy bodies (LBs) composed of abnormally aggregated α-Syn and the progressive reduction of dopaminergic neurons in the substantia nigra pars compacta (SNpc) of the basal ganglia. This causes a lack of dopamine and dysfunction of the entire basal ganglia structure, including the globus pallidus, which is partly responsible for regulating motor and non-motor functions [40,41]. Furthermore, there is also an association between PD and the disruption of calcium homeostasis. Ca^2+^ regulation is crucial for neuronal survival, differentiation, exocytosis at synapses, gene transcription, and proliferation. However, disturbances in calcium homeostasis have been correlated with the selective degeneration of dopaminergic neurons within the SNpc and the promotion of excitotoxicity mediated by this ion, causing oxidative stress of dopaminergic neurons to occur [14,42,43].

HD is considered an inherited neurodegenerative disease caused by the abnormal expansion of cytosine, adenine, and guanine trinucleotide repeats (CAG) in the Huntington gene (HTT), which is located on chromosome 4p16, leading to an abnormally long expression of polyglutamine in the HTT protein and neurodegeneration, which leads to motor, cognitive, and psychiatric symptoms. The progression of the disease causes problems with speech, swallowing, and general functional independence. The epidemiology of HD is considered rare, that is, 5 to 10 cases per 100,000 individuals in most European countries, South America, North America, and Australia, commonly affecting patients between 30 and 50 years old. However, the longer the CAG repeats, the earlier the onset of symptoms. The neuropathological hallmark of HD includes the selective vulnerability of medium spiny neurons (MSNs), which are GABAergic output neurons, and the presence of intracellular aggregates of mutant huntingtin protein (mHTT), which causes neuronal toxicity [44,45,46]. In neurons, mHTT forms intracellular nuclear inclusions and cytosolic aggregates, causing problems with transcription and energy production. In this way, oxidative damage occurs, causing the death of vulnerable cells, mainly medium spiny GABAergic neurons in the dorsal striatum located in the caudate nucleus and putamen, causing atrophy of the affected brain regions such as striatum nucleus, cortical areas, and substantia nigra [47]. Additional features of HD include the degeneration of neurons in the putamen, caudate nucleus, and the cerebral cortex, preferably of enkephalin-containing medium spiny neurons, which are closely related to the onset of chorea, that is, arrhythmic, rapid, and abrupt involuntary movements. Furthermore, loss of substance-P-containing neurons results in dystonia (uncontrolled muscle contractions) and akinesia [48,49]. Finally, the aggressive mechanisms in HD are oxidative stress, neuroinflammation, excitotoxicity, alteration of vesicle trafficking, dysfunction of the organelle clearance system and damaged proteins, reduction of neurotrophin levels, mitochondrial impairment, alteration of the nuclear pore, alteration of DNA methylation, and transcriptional dysregulation [50].

Persistent neuroinflammation after chronic neuropathic pain (NP) may contribute to structural modifications in the brain and lead to mood changes such as depression. Specific inflammatory mediators such as interleukin-1 beta (IL-1β) and tumor necrosis factor-alpha (TNF-α) can influence this. In addition, the pathogenesis of depression is linked to brain-derived neurotrophic factor (BDNF) signaling and impaired synaptic plasticity since BDNF, when associated with oxidative stress through interaction with ROS, triggers neuropsychological disorders [26,51,52].

Neuronal apoptosis and blood–brain barrier (BBB) destruction are the hallmark events of early brain injury after SAH, which are characterized as acute irreversible brain injuries. Increased BBB permeability allows immune molecules to migrate into the brain parenchyma, further aggravating brain injury. Therefore, the inhibition of BBB dysfunction can effectively improve early brain injury after SAH, improving the prognosis of patients with SAH [53,54].

## 3. Phytocompounds

Some phytocompounds have been considered in treating neurodegenerative diseases due to their ability to reduce inflammation and oxidative stress (Table 1).

### 3.1. Alkaloids

#### 3.1.1. Berberine

Berberine is a quaternary botanical isoquinoline alkaloid found in the bark and roots of numerous plants such as *Berberis vulgaris*, *B. aristotle*, *B. aquifolium*, *Hydrastus canadensis*, *Pellodendron chenins*, and *Coptis* rhizomes. This compound has various pharmacological effects, such as anti-obesity, antidiabetic, and antibacterial effects against multiple microbiotas, including many bacterial species, protozoa, plasmodia, fungi, and trypanosomes. In addition, berberine exhibits neuroprotective properties in several neurodegenerative and neuropsychological diseases and antioxidant properties, i.e., it eliminates free radicals, increases endogenous antioxidants, and decreases peroxidative reactions [118,119]. It consists of a crystalline yellow isoquinoline alkaloid traditionally used in Chinese medicine. It has broad therapeutic potential due to its action against various diseases, such as diabetes, hypertension, depression, obesity, inflammation, and cancer [120]. The first information on the medical use of berberine dates back to 200 A.D. However, its low bioavailability limits its application [121].

By being able to penetrate the BBB, berberine can trigger neuroprotection in inflammatory disorders such as, for example, attenuation of neuroinflammation in ischemic and hemorrhagic stroke, spinal cord injury, and TBI. Furthermore, berberine can promote axon regeneration and remyelination in the peripheral nervous system. This is because berberine is closely linked to the inhibition of M1-polarized macrophages and the promotion of M2-polarized macrophages, in addition to causing the inactivation of the nucleotide-binding oligomerization domain-containing protein 3 (NLRP3) inflammasome. The NLRP3 inflammasome is implicated in various central nervous system inflammatory responses, such as cerebral ischemia/reperfusion injury, memory, cognitive impairment, and depression. NLRP3 has also been recently reported to be involved in peripheral nerve inflammation in diabetes-related injuries, NP, and neurodegenerative diseases. The dynamic conversion of macrophage M1 to M2 regulates nerve regeneration after peripheral nerve injury (PNI), and neutrophils and Schwann cells can compensate for this loss of M1 by removing dendrites. M2, in turn, plays dominant pro-regenerative roles after PNI [122].

Berberine inhibits Aβ-induced microglial activity by modulating the suppressor of cytokine signaling 1 (SOCS 1). Through this, it is clear that berberine has a central action in the pathophysiology of AD. In addition, berberine improves endoplasmic reticulum stress and, therefore, attenuates cognitive deficits since endoplasmic reticulum stress is central to signaling by the mechanism of tau protein phosphorylation by hyperactivation of glycogen synthase kinase 3β (GSK 3β) and phosphorylation of eukaryotic translation initiation factor-2α (eIF2α) by activation of PRKR-like endoplasmic reticulum kinase (PERK). By inhibiting this metabolic pathway, berberine may act to reduce brain complications [122,123,124].

In addition, the gut flora contributes to the maturation of microglia, the development of the BBB, and the proliferation and neurogenesis of neurons, revealing the connection of the gut–brain axis. The central nervous system can regulate gut function through direct neural pathways and the hypothalamic–pituitary–adrenal axis, interacting with gut microbes. In contrast, the gut microbiota acts on the central nervous system through neuroendocrine signals and neurotransmitters. In this sense, berberine, besides reducing intestinal inflammation, increasing intestinal permeability, and improving intestinal microbial composition, acts by reducing the accumulation of Aβ plaques, causing less impairment of brain metabolism and reducing the chances of AD [118,125,126].

#### 3.1.2. Huperzine A

Huperzine A is an alkaloid extracted from the Chinese moss *Huperzia serrata*, a natural inhibitor of acetylcholinesterase (AChE) and, therefore, can be used to treat different types of neurodegenerative disorders such as AD and even other problems including bruises, strains, swelling, rheumatism, schizophrenia, myasthenia gravis, and fever [127,128]. It has numerous essential characteristics in brain health. This compound can reduce the level of apoptosis of retinal nerve cells, which is critical in the treatment of glaucoma, and improve the level of serum deprivation and apoptosis mediated by Aβ peptide fragments in cortical neurons. However, huperzine A can also penetrate the BBB, reducing BBB dysfunction and improving age-related neurovascular damage changes. In this way, it is possible to consider the reduction of penetration of immune cells into the brain parenchyma, reducing possible post-SAH side effects [129,130,131].

#### 3.1.3. Caffeine

Coffee beans contain trigonelline, diterpenes, soluble fibers, phenolic compounds, and caffeine, among other bioactive compounds. Caffeine is a xanthine alkaloid compound, characterized by being a powerful stimulant of the nervous system, causing feelings of excitement and alertness, resulting in increased alertness and improved mood. Furthermore, it impacts peripheral tissues such as the heart, skeletal muscle, and adipocytes [132]. Caffeine is also used to prepare ergogenic supplements, as it promotes more significant substrate use, delays fatigue, and enables staying awake. It can also be used in analgesic adjuvant therapies to relieve symptoms related to pathologies, such as body temperature dysregulation and pain [133].

Numerous studies show caffeine has a neuroprotective effect due to its multiple physical and chemical properties, i.e., rapid absorption and organic distribution, with peak accumulation occurring 20–30 min after consumption. Its mode of action is basically through antagonism of adenosine receptors. Depending on the level of consumption acquired by the patient, the chances of developing neurological disorders such as Alzheimer’s, Parkinson’s, epilepsy, and even schizophrenia will be lower [134,135].

The impact of caffeine on the brain circuit is explained by the combined antagonism of adenosine A1 receptors (A1R) that control basal synaptic transmission and adenosine A2A receptors (A2AR) that control synaptic plasticity. However, the administration of this active ingredient should be cautious; that is, moderate levels of caffeine provide robust neuroprotection in neuropsychiatric diseases, ranging from AD, PD, ischemia, epilepsy/seizures, TBI, diabetic encephalopathy, multiple sclerosis (MS), attention deficit hyperactivity disorder, or depressive-type mood dysfunction [136]. Therefore, at low concentrations (250 µM), it acts as an antagonist of adenosine receptors (ARs) A1R, A2AR, A2BR, and A3R, all expressed in neurons and glial cells, with the most excellent affinity for A2AR, preventing the activation of ARs regulating brain functions such as sleep, learning, cognition, and memory. Thus, caffeine regulates the brain diseases mentioned above [137].

During a longitudinal cohort study with more than 20 years of follow-up, it was found that caffeine, measured in plasma caffeine and its metabolites, reduces the risk of PD, with the neuroprotective effects depending on the patient’s exposure to caffeine. Objective blood markers for caffeine metabolism were measured, showing reduced motor impairment, neuronal death, and dopamine depletion. Caffeine is possibly linked to suppressing the A2AR [138,139,140].

In cases of aluminum (Al) poisoning, caffeine may play an essential role in preventing the progression of neurological damage. This metal can cross the BBB, causing neurotoxicity and the possibility of developing neurodegenerative diseases such as AD and PD. In addition, it can increase oxidative stress in the brain and cause conformational changes in numerous proteins through biochemical transformations, such as protein misfolding, aggregation, or oligomerization, contributing to the pathophysiology of neurodegenerative diseases. Therefore, due to its antioxidant properties, caffeine can act as a barrier in this process, in addition to having numerous ARs in the central nervous system, suppressing precursors of the Aβ protein, in addition to contributing to the improvement of cognitive capacity due to the facilitation of synaptic capacity, contributing to the protection against Alzheimer’s [141,142,143,144].

### 3.2. Terpenoids

#### 3.2.1. Ginkgolides

*Gingko biloba* (GB) is one of the plants whose extracts can be used as an adjuvant in treating AD. GB, which belongs to the *Ginkgoaceae* family, is used to treat numerous health problems due to the abundance of bioactive substances it contains, such as terpenoids (ginkgolides A, B, and C), polyphenols, organic acids and flavonoids (quercetin, kaempferol, and isorhamnetin), which are associated with anti-inflammatory, antioxidant, and antiapoptotic effects. The extract can be applied to treat cerebrovascular and cardiovascular diseases. Ginkgolides present several pharmacological activities, such as acting as specific platelet-activating factor antagonists and selective glycine receptor antagonists. However, the ginkgolide content in native ginkgo plants is very low, and the resources of the native ginkgo plant are limited [145,146].

GB inhibits the secretion of proinflammatory mediators and cytokines (nitric oxide, prostaglandin E2 (PGE2), TNF-α, interleukin 6 (IL-6), and IL-1β), promotes the inhibition of cyclooxygenase-2 (COX-2), 5-lipoxygenase (5-LOX), and nuclear factor kappa-light-chain-enhancer of activated B cells (NF-κB), preventing the production of inflammatory substances, inhibits oxidative stress through the inhibition of peroxiredoxin-1 and, finally, inhibits the expression of adhesion factors like intercellular adhesion molecule 1 (ICAM-1) and vascular cell adhesion molecule 1 (VCAM-1), which are necessary for the adhesion of leukocytes to the surface of blood vessels and subsequent migration to inflammation. In this way, it acts insistently to prevent neuroinflammation [147,148,149].

#### 3.2.2. Limonene

One of the most common terpenes in nature, limonene, can be found in citrus peel oil, dill oil, caraway oil, neroli, bergamot, and cumin. This phytochemical has numerous biochemical effects, mainly anticancer, as it can induce apoptosis through the positive regulation of pro-apoptotic factors and the negative regulation of antiapoptotic factors [150].

Limonene is a monocyclic monoterpene found in citrus peel oils. It has a protective effect against neurological disorders such as AD, stroke, and cerebral ischemia through its antioxidant, anti-inflammatory, and anti-cancer neuroprotective properties. More specifically, limonene can attenuate locomotor deficits in PD by inhibiting rotenone (ROT), which, when expressed, induces the inhibition of complex-1 of the mitochondrial electron transport chain, causing a syndrome replicating PD’s neuropathological and behavioral symptoms. In addition, it prevents abnormal protein deposition, prevents damage to mitochondrial function, and induces apoptosis of abnormal cells [151,152,153].

Figure 1 shows the main effects of ginkgolides, curcumin, and limonene. As demonstrated anteriorly, ginkgolides inhibit several signaling pathways associated with neuroinflammation and synaptic dysfunctions appropriately to counteract neurodegeneration. Moreover, limonene exerts beneficial effects against protein deposition and neuronal dysfunctions. Curcumin is a potent anti-inflammatory and antioxidant component that limits neuroinflammation and neuronal oxidation. These are crucial aspects related to neurodegenerative disease occurrence.

### 3.3. Polyphenols

#### 3.3.1. Rosmarinic Acid

Rosmarinic acid is a polyphenol compound found in several species of the *Lamiaceae* family, and it has anti-inflammatory, antioxidant, and neuroprotective properties. As a natural product, its consumption can promote neuroprotection by activating anti-inflammatory and antioxidant signaling pathways, improving cognitive functions. In addition, this compound helps in the homeostasis of glucose, triglycerides, free fatty acids, and cholesterol, preventing their accumulation in the body, especially in nerve pathways. Finally, within nerve synapses, rosmarinic acid also acts on brain plasticity, neurogenesis, and hippocampal memory consolidation [154].

Rosmarinic acid is a phenolic acid commonly found in plants from the *Lamiaceae* family, such as *Perilla frutescens* (L.) Britton, *Rosmarinus officinalis* L., and *Melissa officinalis* L., which are used in teas, herbs, culinary condiments, spices, and fruits [155]. This compound has interesting nutritional properties, mainly because it is a potent antioxidant and can help prevent and treat tumors. Among the active antitumor components, polyphenols stand out, inhibiting tumor growth and enhancing chemo-radiotherapy agents. However, the compound is generated through purification after plant biosynthesis and has a low bioavailability [156].

In addition, rosmarinic acid plays a considerable role in combating AD by suppressing Aβ aggregation. Furthermore, cognitive decline is likely due to suppression of tau phosphorylation since the accumulation of this phosphorylated protein impairs hippocampal neurogenesis by suppressing GABAergic transmission. [157,158].

In PD, rosmarinic acid acts at different levels to prevent its progression. Typically, PD patients with cognitive impairment have elevated levels of phosphorylated Tau along with α-Syn. In addition, disruption of mitochondrial biogenesis is associated with the development of PD. However, romarinic acid markedly reduces α-Syn expression and Tau phosphorylation. In addition, it activates the antioxidant enzyme HO-1 to suppress the production of ROS and cell death induced by H_2_O_2_ exposure. Another neuroprotective factor of rosmarinic acid is restoring mitochondrial complex I function and recovering dopamine content, attenuating PD [159].

#### 3.3.2. Ferulic Acid

Ferulic acid (4-hydroxy-3-methoxycinnamic acid) is found in whole grains, grapes, parsley, spinach, cereal seeds, artichokes, coffee, and cereal seeds, particularly wheat, oats, rye, and barley. It has antioxidant, anti-inflammatory, antibacterial, antithrombotic, antitumor, and neuroprotective activity [160,161,162]. Its importance is due to its antioxidant activity (neutralizes free radicals in muscle tissue, relieving muscle fatigue), low toxicity, and other physiological functions, including anti-inflammatory, antimicrobial, anti-cancer activity (for example, lung, breast, colon, and skin cancer), antiarrhythmic, and antithrombotic effects. Ferulic acid also possesses antidiabetic effects and immune-stimulating properties, reduces damage to nerve cells, and can aid in the repair of damaged cells. It has been widely used in pharmaceuticals and foods. However, its use is limited by its tendency to be rapidly oxidized in the body [163].

Ferulic acid may also prevent cerebral ischemia/reperfusion injury and is neuroprotective. It is believed that the neuroprotective activity against neuroinflammation occurs through the inactivation of the expression of the NRLP3 domain, causing the mitigation of cellular damage, mainly in the vascular endothelium, reducing the chances of thromboembolisms and ischemia [164,165].

Ferulic acid also protects against depression. Clinical evidence suggests that inflammation mediated by IL-6, IL-1β, and neurotrophins such as BDNF and nerve growth factor (NGF) alters the metabolism of the essential amino acid tryptophan (Trp). Trp is the precursor for the synthesis of 5-hydroxytryptamine (5-HT) and plays a critical role in the pathobiology of depression [166,167]. However, ferulic acid inhibits these cytokines and oxidative stress, mitigating depressive symptoms [168,169].

#### 3.3.3. Caffeic Acid

Caffeic acid is an organic compound with two functional groups (phenolic hydroxyl and acrylic acid). It is widely distributed in medicinal plants, vegetables, and fruits and exhibits several pharmacological properties, such as antioxidants, antibacterials, antivirals, antitumor, anti-inflammatory, and neuroprotective properties. It also regulates blood glucose and lipids [170]. However, extracting this phytochemical is complicated, as it is affected by the plant’s growth cycle, climatic environment, and other factors.

AD consists of accumulating Aβ peptide and forming neurofibrillary tangles in the brain. Aβ production occurs through the action of β-secretase, also known as β-site amyloid precursor protein cleaving enzyme-1 (BACE-1), generating AD-associated toxicity and neurodegeneration, while neurofibrillary tangles, formed from tau hyperphosphorylation, prevent the stabilization of neuronal microtubules and disrupt the pathway for intracellular transport. These two substances accumulated in the brain cause oxidative stress, neuroinflammation, downregulation of brain neurotrophic factors, and synaptic dysfunction. Thus, caffeic acid prevents oxidative stress by inhibiting inflammatory factors like NF-κB. Furthermore, it stimulates the endogenous cellular antioxidant mechanism, such as the Nrf2 signaling pathway. Caffeic acid also increased the expression of synaptic proteins such as the phosphatidylinositol 3-kinase/protein kinase B (PI3K/Akt), improving hippocampal plasticity and memory functions [171,172,173].

#### 3.3.4. Resveratrol

Resveratrol (3,5,4′-trihydroxystilbene) is an estrogen-like phytosterol found in grapes, peanuts, blueberries, and other foods. This compound has excellent protective potential in the nervous system by mediating blood pressure and lipid profiles, which are key factors in managing and preventing stroke [174]. Resveratrol can be extracted from over 70 plants, including peanuts, pistachios, blueberries, blackberries, and grapes, whose skin is particularly rich in resveratrol. Plants produce this substance to protect them from environmental stress and pathogenic invasion. It has cardioprotective, antioxidant, anti-inflammatory, antiaging, and tumorigenesis preventive effects [175].

Resveratrol has a direct antioxidant contribution to tissues, especially the brain, even though its concentration is low when administered orally. Its action occurs through the 67 kDa laminin receptor (67LR) located on the cell surface and mediates the neuroprotective actions of resveratrol [176]. In AD, the accumulation of Aβ plaques will lead to the formation of hyperphosphorylated tau protein as neurofibrillary tangles, vascular damage, and cell death, eventually resulting in dementia. However, resveratrol acts by inhibiting the deposition of Aβ proteins and the formation of proinflammatory cytokines (TNF-α, IL-1β, and IL-6), protecting against the progression of AD [177,178].

#### 3.3.5. Pterostilbene

Pterostilbene (3′,5′-dimethoxy-resveratrol) is a plant secondary metabolite initially isolated from the heartwood of red sandalwood (*Ptocarpus santalinus*). However, it is also found in other plants and berries, such as blueberries. This compound is involved in the plant’s defense against stressful conditions, i.e., ultra-violet (UV) radiation, pathogen aggression, low soil fertility, high/low temperatures, or severe drought [179]. It has a notable action in preventing inflammatory dermatoses, promoting photoprotection, cancer prevention, and therapy, improving insulin sensitivity, decreasing lipids and glucose levels, protecting against cardiovascular diseases, and promoting improvements in memory and cognition [180].

Pterostilbene exhibits resistance to metabolic modifications, allowing a more significant fraction to reach its target sites and remain unchanged for longer than resveratrol. Therefore, it is considered a more efficient and promising compound. It has neuroprotective, antioxidant, anti-inflammatory, antidiabetic type 2, antiapoptotic, and antiaging activities. In addition, this compound can reduce AChE activity, improve antioxidant parameters, and enhance cholinergic neurotransmission, possibly exhibiting an effect against AD [181].

Neuroinflammation is the critical process in the worsening of secondary injury after intracerebral hemorrhage, leading to a poor prognosis, in addition to inflammation aggravating BBB damage, neuronal death, and neurological impairment. After intracerebral hemorrhage, microglia are activated by releasing inflammatory cytokines into the extracellular environment to trigger an inflammatory reaction exacerbating secondary brain injury. However, pterostilbene can cross the BBB, inhibiting inflammatory and mitochondrial oxidative stress injury after cerebral ischemic stroke and reducing inflammation and early brain injury involved in oxidation after SAH. The mechanism is that pterostilbene suppresses microglial activation through multiple pathways, including inhibition of the NLRP3/caspase-1 inflammasome pathway, activation of the silent mating type 2 homolog 1 (SIRT1) signaling pathway, and inhibition of the NF-κB signaling pathway [182,183,184].

#### 3.3.6. Luteolin

Luteolin is a natural flavonoid widely found in many plant species. It is mainly present in fruits and vegetables such as celery, chrysanthemum flowers, sweet peppers, carrots, onion leaves, broccoli, and parsley. Luteolin presents several health benefits, and this compound can be used as an effective strategy as an adjuvant in treating or preventing many human diseases, including hypertension, inflammatory disorders, and, ultimately, cancer [185,186,187,188,189]. It (3′,4′,5,7-tetrahydroxyflavone; C_15_H_10_O_6_) belongs to a family of natural secondary metabolites called flavonoids characterized by a diphenylpropane (C_6_-C_3_-C_6_) structure, more specifically to the subgroup of flavonoids categorized as flavones. It is present in vegetables of the *Apiaceae* family, such as carrots, dried parsley, broccoli, green pepper, spinach, and cabbage [1,190].

It has anti-inflammatory, antioxidant, anti-apoptotic, and neuroprotective activities. This compound acts by regulating the activity of peroxisome proliferator-activated receptor gamma (PPARγ). This essential nuclear receptor regulates the transcription of all types of genes involved in cell differentiation, inflammation, oxidative stress, lipid metabolism, and glucose homeostasis. It is known that the expression of γ-secretase and β-secretase can be inhibited through the binding of PPARγ to the promoter region of the BACE1 gene, subsequently suppressing the generation of Aβ since Aβ peptides are derived from the sequential break of the amyloid protein precursor (APP), an essential single transmembrane protein, by BACE1, providing the possible development of AD. Aβ deposition causes oxidative stress, endoplasmic reticulum stress, mitochondrial dysfunction, and neuronal apoptosis, eventually leading to cognitive impairment [191,192].

Luteolin also prevents excessive mitophagy and autophagy through mammalian rapamycin (mTOR) signaling targets. A subgroup of the mTOR family, the mammalian target of rapamycin complex 1 (mTORC1), regulates metabolism, cell growth, and autophagy. Under normal physiological conditions, mTORC1 promotes cell growth and protein synthesis and hinders autophagy by phosphorylating Unc-51 Like Autophagy Activating Kinase 1 (ULK1). However, mTORC1 activity is suppressed under stress conditions. Thus, mTORC1 regulation is essential for maintaining cellular homeostasis and preventing uncontrolled autophagy. This complex process can suppress excessive cellular degradation even in abnormal levels of glutamate in the extracellular space of the central nervous system. This process occurs in Alzheimer’s, Parkinson’s, and Huntington’s [193,194].

The pathophysiology of HD is mediated by an imbalance in ROS, which can lead to mitochondrial damage and neuron death. Luteolin, however, induced the expression of the Nrf2/HO-1 antioxidant pathway, i.e., through this metabolic pathway, oxidative stress was inhibited, reducing neurodegeneration in HD. Nrf2 is released from kelch-like ECH-associated protein 1 (Keap1), escaping proteasomal degradation and translocating to the nucleus. Nfr2 binds to the antioxidant response element (ARE) in the nucleus. It initiates the transcription of a series of antioxidant enzymes, such as HO-1, glutathione S-transferase (GST), SOD, CAT, glutathione reductase, and nicotinamide adenine dinucleotide phosphate (NAD(P)H). These enzymes reduce cellular oxidative stress and the formation of free radicals [195,196].

Due to its free radical degradation function, luteolin can suppress lipid peroxidation and restore antioxidant defense. It also prevents gene expression of proinflammatory factors in microglia, resulting in unique anti-inflammatory and neuroprotective properties in certain diseases like depression [197,198,199].

Another mechanism luteolin exerts is its activity against brain aging through its action on D-galactose (D-gal). In an experiment with rats, the action of luteolin on behavior, cholinergic functions, and mitochondrial respiration of the hippocampus were evaluated, taking into account gene expressions of SIRT1, BDNF, and receptors for advanced glycation end products (RAGE). Finally, luteolin alleviated D-gal-induced cognitive impairment, reversed cholinergic abnormalities, and reduced hippocampal oxidative stress, mitochondrial dysfunction, neuroinflammation, brain senescence, and neural apoptosis [200].

Another condition that luteolin can fit in is glioblastoma due to its antioxidant and anti-inflammatory actions, which have shown promise in inhibiting cancer cell growth, promoting apoptosis, and tremendous capacity to modulate growth pathways of this cancer [201].

Another function of luteolin is that it has modulatory effects on biochemical signaling pathways associated with endogenous antioxidant systems, increased mitochondrial functions, and inhibition of neuroinflammation by modulating signaling pathways such as NF-κB, the Janus kinase pathway and the signal transducer and activator of transcription (JAK/STAT) protein, the Toll-like receptor (TLR) pathway, and the cAMP response element-binding protein (CREB) pathway, which are involved in neuroinflammation associated with significant neurological and psychiatric disorders, including autism spectrum disorder [202,203].

#### 3.3.7. Quercetin

Quercetin (3,3′,4′,5,7-pentahydroxyflavone) is categorized as a flavonol, one of the six subclasses of flavonoid compounds. The name has been used since 1857 and is derived from quercetum (oak forest). It can be found in fruits (apples, berries, capers, grapes, onions, and tomatoes) and seeds such as nuts. Quercetin possesses unique biological properties that can improve mental and physical performance and reduce the risk of infection. It exhibits anti-cancer, anti-inflammatory, antiviral, antioxidant, and psychostimulant activities, along with the ability to inhibit lipid peroxidation, platelet aggregation, and capillary permeability, as well as stimulating mitochondrial biogenesis [204,205]. Moreover, it can exhibit anti-apoptotic, neuroprotective, and hepatoprotective effects [206].

Cerebral ischemia is known to cause inactivation of physiological pathways, initiating a series of biochemical events such as excitotoxicity, intracellular calcium overload, oxidative damage leading to microvascular injury, dysfunctions of the BBB, an inflammatory cascade involving TNF-α and NF-κB, and ultimately the activation of pro-apoptotic genes such as P38 and c-Jun N-terminal kinases (JNK), resulting in permanent and irreversible neuronal loss. Specific brain regions are more affected by the cessation of nutritional supply during ischemic episodes, including the cortex, striatal nucleus, thalamus, hypothalamus, and hippocampus. However, quercetin has a significant protective effect in ischemic conditions, as it reduces the rate of cell apoptosis and exhibits antioxidant and anti-inflammatory activities, key mechanisms following ischemia. Furthermore, the amphiphilic properties of quercetin facilitate its rapid passage across the BBB, allowing for a wide range of interactions with brain proteins [207,208].

In MS, quercetin also plays a significant role. It is known that the reduction of oligodendrocytes, destruction of myelin, and glial activation are essential factors in the pathophysiology of MS, as the differentiation and recruitment of oligodendrocyte progenitor cells (OPCs) are often insufficient to overcome demyelination. However, quercetin protects OPCs, reduces apoptosis, and enhances their proliferation and differentiation. The PI3K/Akt signaling pathway can explain this action. Additionally, quercetin reduces the activation of glial cells (astrocytes and microglia) by preventing the release of inflammatory factors, thus inhibiting processes such as astrogliosis and microgliosis. Finally, this phytochemical also reduces astrocyte activation by inhibiting the transition from the G1 phase to the S phase of the cell cycle and preventing astrocyte proliferation by blocking the extracellular signal-regulated kinase (ERK)/focal adhesion kinase (FAK) pathway [209,210].

In AD, quercetin acts on the heat shock protein beta-1 (HSPB1), the redox-sensitive transcription factor Nrf2, and the neurotrophins receptor tropomyosin receptor kinase B (TRKB), inhibiting Tau aggregation and protecting differentiated neuronal cells from the neurotoxicity of hyperphosphorylated Tau [211]. Additionally, quercetin reduces the production of Aβ plaques, the primary substance involved in AD pathology [212].

Figure 2 summarizes the effects of flavonoids and alkaloids on neurodegenerative diseases. As anteriorly demonstrated, flavonoids and alkaloids present several neuroprotective effects related to the fight against neurodegeneration. Being anti-inflammatory, antioxidant, and anti-apoptotic may be a sufficient strategy to counteract neuroinflammation, neuronal oxidative stress, and synaptic dysfunctions, which are crucial neurodegeneration occurrences [213].

#### 3.3.8. Curcumin

*Curcuma longa* is part of the ginger family (*Zingiberaceae*), originating in India and currently cultivated in many countries, including Southeast Asia, China, and Latin America. It is a spice commonly used to produce different curries in India and other Asian countries due to its flavor and color. India is the largest producer and leading exporter. Turmeric has several medicinal properties and is therefore used in dietary supplements, which can help with joint comfort, promote mobility and flexibility, and improve cognitive functioning and cardiovascular health [214,215].

While ROS are essential for signaling under normal physiological conditions through signal transduction and immune regulation, their excess is associated with many central nervous system diseases, such as TBI, spinal cord injury, AD, and PD. Their accumulation can induce severe oxidative stress, i.e., it causes oxidative damage to DNA, lipids, and proteins, generating widespread secondary lesions, such as axonal demyelination and neuronal cell necrosis. However, the balance of endogenous ROS is maintained by an antioxidant protection system comprising oxidase (OXD), peroxidase (POD), SOD, and CAT, through the catalysis of the conversion of superoxide anion (•O^2^) into H_2_O_2_ and oxygen (O^2^). In this scenario, curcumin can inhibit the inflammatory response by repolarizing microglia and macrophages, reducing ROS formation, and activating the Nrf2/HO-1 signaling pathway, which also eliminates free radicals and positively modulates the regulation of SIRT1 information, which prevents neurodegeneration, oxidative stress, and neuroinflammation [216,217,218,219].

Figure 3 summarizes the effects of polyphenols on neurodegenerative diseases. Table 1 summarizes the main phytocompounds and their main neuroprotection effects, and Table 2 shows different clinical trials that evaluated the impact of phytocompounds on neurodegenerative conditions. In summary, these clinical trials show that the use of these compounds is related to significant improvements in cognition, memory, disinhibition, irritability/lability, aberrant behavior, hallucinations, and depression.

## 4. Phytocompounds’ Effects on Neurodegeneration: A Comprehensive Overview Based on Clinical Trials

Table 2 comprehensively overviews how phytochemicals impact neurodegeneration through various pathways and presents a comparative analysis of multiple interventions and outcomes.

### 4.1. Literature Search Methodology

#### 4.1.1. Focal Question

The focal question considered in our review was “What are the effects of phytochemicals against neurodegeneration?”.

#### 4.1.2. Language

We included only studies that were published in English.

#### 4.1.3. Databases

We consulted PubMed, EMBASE, and COCHRANE databases to build this review. The mesh terms were “Phytochemicals”, “Neurodegeneration”, “Alzheimer’s disease”, “Parkinson’s disease”, “Neuroinflammation”, “Oxidative stress”, “Inflammation”, “Pro-inflammatory”, “Molecular pathways”, “Terpenoids”, “Alkaloids”, and “Polyphenols”. The mesh terms enabled the authors to search and identify relevant clinical studies related to this review’s objectives. The Preferred Reporting Items for a Systematic Review and Meta-Analysis (PRISMA) guidelines were strictly followed to build this review with the utmost transparency and rigor [233].

#### 4.1.4. Study Selection

For this comprehensive review, we included randomized clinical trials and non-randomized clinical trials that investigated the effects of terpenoids, alkaloids, and polyphenols on conditions related to neurodegeneration. The inclusion criteria included double-blind, placebo-controlled, and open-label studies. Only full texts were considered. The exclusion criteria were preclinical studies (animal models and in vitro studies), studies published in languages rather than English, reviews, meta-analyses, poster presentations, case reports, and editorials.

#### 4.1.5. Data Extraction

We did not restrict the period for searching for randomized and non-randomized clinical trials. Table 2 lists the included studies.

### 4.2. Literature Search Report

The PRISMA Flow Diagram below (Figure 4) demonstrates the literature search report. Following the review’s initial phase, 150 studies were identified from reputable databases, including PubMed, and 13 records were sourced from registers. After compiling these records, 98 duplicate records were eliminated, followed by 35 records marked as ineligible by automation tools, and 12 were removed for other reasons, which could include issues such as incomplete data or irrelevant content. After eliminating these records, 18 remained for the screening process, and five were excluded based on their content, relevance, or quality unrelated to this review’s scope. The remaining 13 reports were then sought for retrieval, and all 13 reports were successfully retrieved. No further reports were excluded, resulting in 13 studies being included in the review. Importantly, no registerswere included.

### 4.3. Potential Therapeutic Effects of Phytochemicals in Neurodegeneration: Current Findings

Nakamura et al. [220] conducted a double-blind, randomized, placebo-controlled study to evaluate quercetin’s potential neuroprotective and cognitive enhancer effects in eighty healthy individuals. Their study highlighted that 500 mL of barley tea containing 110 mg of quercetin effectively enhanced cognitive function and flexibility, improving mental and motor actions. The authors highlighted quercetin’s potential in modulating amyloid protein depositions within the study’s patients. However, they did not evaluate cerebral protein deposition using any method. Moreover, quercetin was well tolerated, and no potential adverse events were observed.

Another study examined the effects of huperzine A in double-blind controlled studies. These studies utilized Alzheimer’s patients and healthy individuals. Their results demonstrated that patients receiving 0.2 mg daily of huperzine A for two weeks significantly improved cognitive domains and task-switching skills compared to those not receiving the supplement. However, considering the long-term occurrence of Alzheimer’s, it is potentially problematic to draw conclusions based on such a small intervention period. Moreover, the intervention did not introduce severe side effects and was well tolerated [221,222].

Our literature search analysis presented three studies examining the effects of caffeine as neuroprotective. These authors utilized healthy individuals and discovered that oral caffeine ingestion positively affected attention, cognitive control, and memory. The results demonstrated positive effects on visual and auditory memory, highlighting the potential stimulatory effects of caffeine on decreasing synaptic dysfunctions [223,224,225]. The included studies did not encounter significant adverse effects from their interventions. However, future studies might be necessary with diseased patients rather than healthy individuals to potentialize the findings into the elderly population, which is the most affected by neurodegeneration.

Dodge et al. [226] experienced ginkgolides and found that the effects of the phytochemicals were neuroprotection. Their results demonstrated sufficient and significant protection against depression and dementia as the intervention group received 240 mg daily (80 mg, 3 times daily) for 42 months. This extended treatment period is of utmost importance not only to address the long-term effects of the intervention but also to demonstrate the adverse events of the included treatment. However, the authors did not find any severe adverse events. Future research must endeavor to find adaptable treatment strategies, diminishing the number of doses daily since the included sample comprised three dosages per day, which might decrease patients’ adhesion to the proposed treatment.

Kuszewski et al. and Rainey-Smith et al. [227,228] explored curcumin, the major phytochemical from *Curcuma longa*, against cognitive decline. Subjects received a single daily dose of 400 mg for 12 months, and the results demonstrated potent cognitive enhancement. Future research must adhere to the intervention against Alzheimer’s or Parkinson’s, which are related to neuroinflammation since curcumin is a powerful anti-inflammatory compound. The included authors did not receive any serious adverse events.

On the other hand, Noguchi-Shinohara et al. [229,230] examined the effects of rosmarinic acid against severe dementia in a randomized, placebo-controlled, double-blind clinical trial. Patients received 500 mg of rosmarinic acid daily for 24 weeks, and the results demonstrated a tremendous positive effect on agitation, therefore being significant without serious adverse events. However, further research must examine the effects of rosmarinic acid against dementia based on its signaling pathway modulations, thus integrating findings and generalizing results to a broader range of diseases sharing pathophysiological similarities.

Kimura et al. [231] studied ferulic acid, a bioactive compound, against depression symptoms among 20 patients. Their results demonstrated significant improvements in disinhibition, irritability/lability, aberrant behavior, hallucinations, and depression among the intervened individuals without any serious events.

Lastly, Witte et al. evaluated resveratrol, which is richly found in grapes, against memory dysfunctions in elderly individuals. The results demonstrated that following a daily intake of four capsules (200 mg of resveratrol) over 26 weeks significantly and positively affected memory functions, especially in word retention. The results were without serious adverse effects, and future research must generalize the findings based on the signaling pathways involved in translating the results into diagnostic groups based on the shared pathophysiological pathways [232].

## 5. Conclusions and Future Research Perspectives

The role of phytocompounds such as polyphenols, alkaloids, and terpenoids in the prevention or treatment of neurodegenerative diseases is based mainly on antioxidant, anti-inflammatory, and inhibitory properties on certain neurotransmitter-degrading enzymes, causing an increase in the concentration and activity of other types of neurotransmitters, reducing neuroinflammation, protecting defense cells such as microglia, delaying the progression of dementia, and improving pro- and anti-apoptotic properties. In humans, the treatment with these compounds can significantly improve cognition, memory, disinhibition, irritability/lability, aberrant behavior, hallucinations, and depression. Furthermore, phytoconstituents cause fewer adverse effects and may have lower costs.

In this scenario, phytochemicals against neurodegeneration may present a new and vital investment opportunity for the pharmaceutical industry [234]. Many medications widely used worldwide are derived from phytochemicals and functional biomolecules. Secondly, phytochemicals can be used as modulators of already existing synthetic anti-neurodegenerative medications, enhancing their effectiveness and reducing potential adverse events associated with their use. The prevention of neurodegeneration is another promising area for pharmaceutical investment. Using phytochemicals as anti-inflammatory, antioxidant, and immunomodulators against neurodegeneration in various predisposing conditions is paramount.

Furthermore, phytochemicals may vary in action between individuals and populations, posing challenges regarding drug responses, toxicities, and resistance [235]. In this scenario, genome-wide association studies (GWAS) may be helpful. GWAS compares genomes of many individuals and populations to identify markers associated with disease or its therapy. Conducting GWAS against neurodegenerative genes and their associations with phytochemical-based treatments against AD, MS, or PD may help compare the effectiveness and combine therapies against neurodegeneration based on personalized strategies, enhancing individual patients’ outcomes. Combining phytochemicals with nanoparticles or extracellular vesicles that are more prominent and capable of passing the BBB may be another great avenue for future research and to improve personalized therapies against neurodegenerative diseases. Nanoparticles often enhance phytochemicals’ solubility, distribution, and bioavailability, therefore targeting cells more efficiently. Exosomes are naturally occurring extracellular vesicles that improve the stability of their intramembrane components. Thus, using these strategies to counteract neurodegeneration in combination with phytochemicals can lead to better dose- and time-dependent effectiveness.

## Figures and Tables

**Figure 1 metabolites-15-00124-f001:**
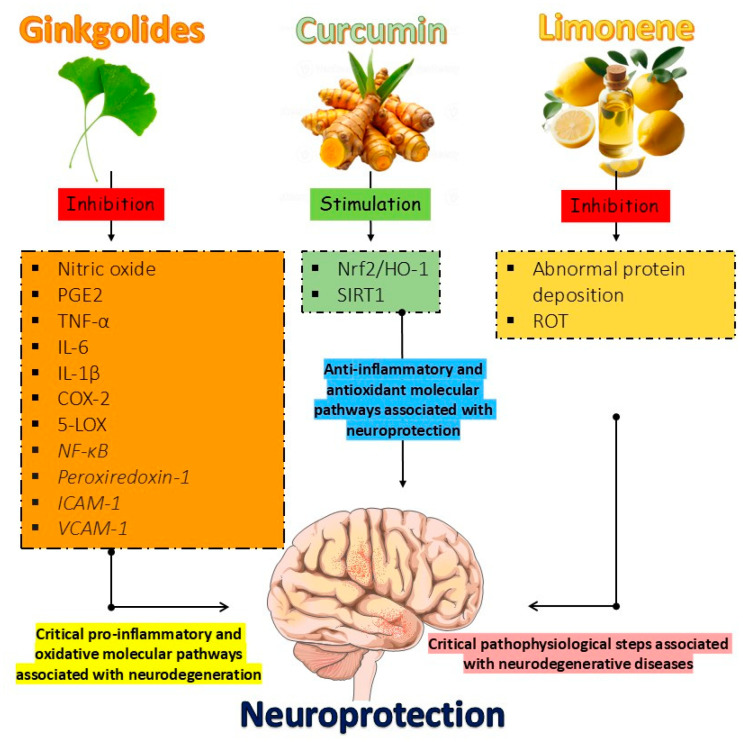
During neurodegeneration, critical pro-inflammatory and oxidative molecular pathways are involved in the occurrence of neurodegenerative conditions. However, protective anti-inflammatory and antioxidant pathways may improve neurofunction. Imbalances between these two sets of events may predispose to protein deposition and diseases related to the brain. Ginkgolides are essential in combating various inflammatory substances such as, for example, nitric oxide, prostaglandin E2 (PGE2), tumor necrosis factor-alpha (TNF-α), Interleukin-6 (IL-6), interleukin-1beta (IL-1β), cyclooxygenase-2 (COX-2), 5-lipoxygenase (5-LOX), and nuclear factor kappa-light-chain-enhancer of activated B cells (NF-κB). Furthermore, it also inhibits adhesion factors such as intercellular adhesion molecule 1 (ICAM-1) and vascular cell adhesion molecule 1 (VCAM-1), reducing inflammatory processes in the brain, as these are essential factors for the action of defense cells. Curcumin, in turn, reduces reactive oxygen species (ROS) formation through the activation of the nuclear factor erythroid-2 related factor 2/heme oxygenase (Nrf2/HO-1) signaling pathway, which eliminates free radicals and positively modulates the regulation of silent mating type 2 homolog 1 (SIRT1) information, preventing neurodegeneration and neuroinflammation. Finally, limonene can attenuate locomotor deficits in Parkinson’s disease (PD) by inhibiting rotenone (ROT), which, when expressed, induces the inhibition of complex-1 of the mitochondrial electron transport chain, causing a syndrome replicating PD’s neuropathological and behavioral symptoms. It also inhibits abnormal protein deposition, preventing mitochondrial function impairments and inducing apoptosis of abnormal cells.

**Figure 2 metabolites-15-00124-f002:**
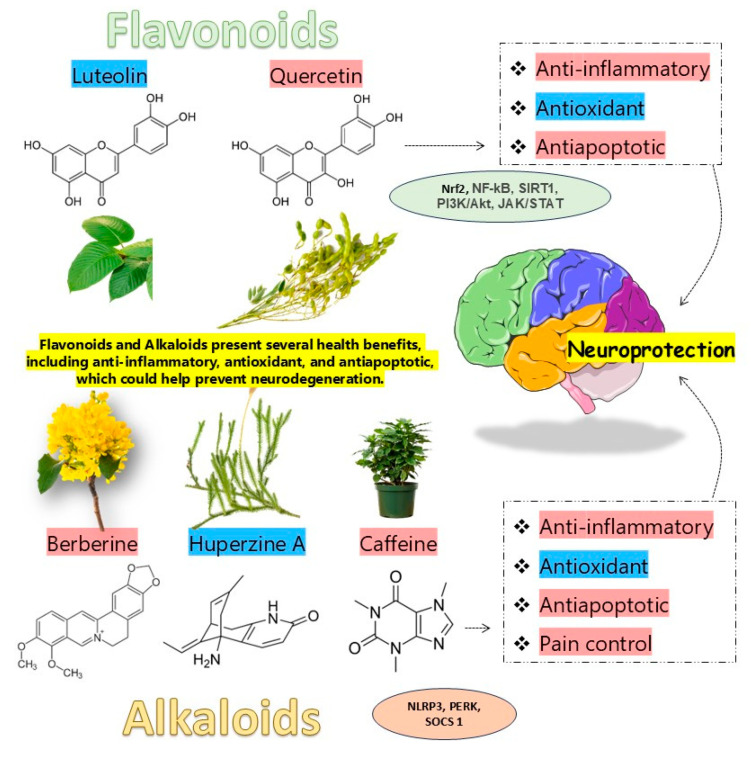
Our manuscript denotes the neuroprotective effects of many phytochemical classes, including those from flavonoids and alkaloids, against neurodegeneration. Flavonoids (luteolin and quercetin) and alkaloids (berberine, huperzine A, and caffeine), as demonstrated by the included studies, exert anti-inflammatory, antioxidant, and anti-apoptotic activities, therefore improving pain control and promoting cognitive stimulation. In addition, alkaloids are neuro-stimulants, thus enhancing neurotransmission. **Abbreviations:** JAK/STAT, Janus kinase pathway and the signal transducer and activator of transcription; NF-κB, nuclear factor kappa-light-chain-enhancer of activated B cells; NLRP3, nucleotide-binding oligomerization domain-containing protein 3; Nrf2, nuclear factor erythroid-2 related factor 2; PERK, PRKR-like endoplasmic reticulum kinase; PI3K/Akt, phosphatidylinositol 3-kinase/protein kinase B; SIRT1, silent mating type 2 homolog 1; SOCS 1, suppressor of cytokine signaling 1.

**Figure 3 metabolites-15-00124-f003:**
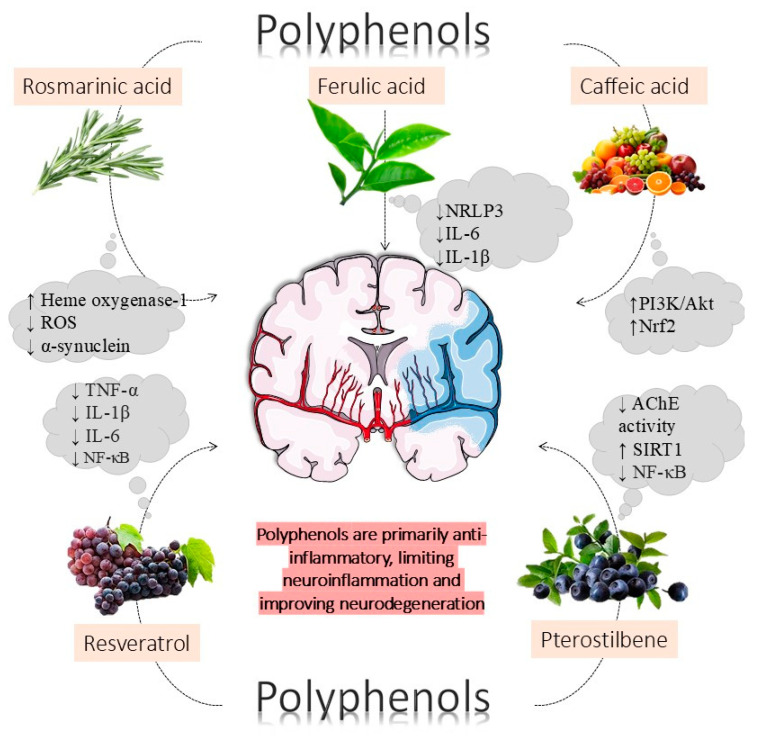
Polyphenols are primarily anti-inflammatory bioactive compounds. They counteract neuroinflammation, therefore improving neurodegeneration and preventing neurodegenerative diseases. Rosmarinic acid combats Parkinson’s disease (PD) by suppressing α-synuclein expression and Tau phosphorylation. In addition, it activates the antioxidant enzyme heme oxygenase-1 to suppress the production of reactive oxygen species (ROS) and promotes the restoration of mitochondrial complex I function and recovery of dopamine content, attenuating PD. Ferulic acid prevents cerebral ischemia/reperfusion injury and anti-neurological diseases through the inactivation of the nucleotide-binding oligomerization domain-containing protein 3 (NRLP3) domain reducing the chances of thromboembolisms and ischemia and also acts in the protection against depression by the suppression of inflammation mediated by interleukin-6 (IL-6), and interleukin-1 beta (IL-1β). Caffeic acid stimulates the endogenous cellular antioxidant mechanism, such as nuclear factor erythroid 2-related factor 2 (Nrf2). Also, it increases the expression of synaptic proteins such as phosphatidylinositol 3-kinase/protein kinase B (PI3K/Akt), improving hippocampal plasticity and memory functions. Resveratrol acts by inhibiting the nuclear factor kappa-light-chain-enhancer of activated B cells (NF-κB) and the deposition of amyloid beta (Aβ) proteins, limiting the formation of proinflammatory cytokines tumor necrosis factor-alpha (TNF-α), IL-1β, and IL-6, and protecting against the progression of Alzheimer’s disease (AD). Finally, pterostilbene can reduce acetylcholinesterase (AChE) activity, improving antioxidant parameters and enhancing cholinergic neurotransmission, and avoid neuroinflammation by suppressing microglial activation through the activation of the silent mating type 2 homolog 1 (SIRT1) signaling pathway and inhibition of the NF-κB signaling pathway.

**Figure 4 metabolites-15-00124-f004:**
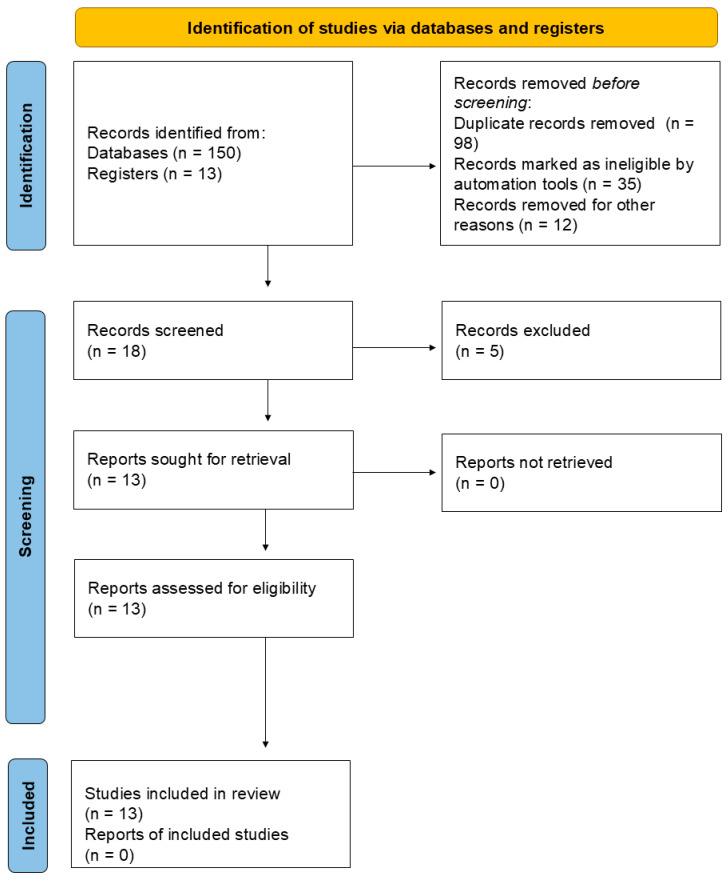
PRISMA flow chart overview.

**Table 1 metabolites-15-00124-t001:** Phytocompounds and functions related to neuroprotection.

Name	Chemical Structure	Neuroprotection Effects	Ref.
Luteolin		Anti-inflammatory, antioxidant, antiapoptotic, antimitotic, and regulation of peroxisome activity.	[55,56,57,58,59]
Quercetin	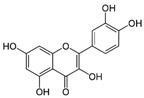	Antioxidant, anti-inflammatory, and antiapoptotic.	[60,61,62,63,64]
Berberine	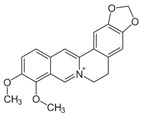	Antioxidant, anti-peroxidative reactions, and anti-inflammatory.	[65,66,67,68,69,70]
Huperzine A	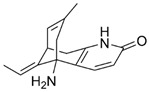	Anti-inflammatory and antiapoptotic.	[71,72,73,74,75]
Caffeine	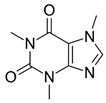	Stimulant, antioxidant, anti-inflammatory, and pain control.	[76,77,78,79,80]
Ginkgolides	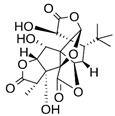	Anti-inflammatory, antioxidant, and antiapoptotic.	[81,82,83,84,85]
Curcumin	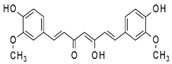	Antioxidant, anti-inflammatory, and antiapoptotic.	[86,87,88,89,90]
Limonene	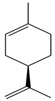	Antioxidant, anti-inflammatory, and anti-cancer.	[91,92,93,94]
Rosmarinic acid	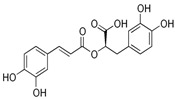	Antioxidant, anti-inflammatory, and improvement of cerebral plasticity and neurogenesis.	[95,96,97]
Ferulic acid	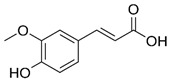	Antioxidant, anti-inflammatory, antibacterial, antithrombotic, and antitumor.	[98,99,100,101,102]
Caffeic acid	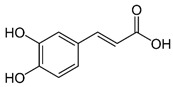	Antioxidant, anti-inflammatory, and promotion of cerebral plasticity.	[103,104,105,106,107]
Resveratrol	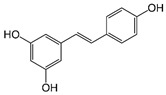	Antioxidant, antiapoptotic, and anti-inflammatory.	[108,109,110,111,112]
Pterostilbene	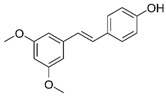	Antioxidant, anti-inflammatory, antiapoptotic, and antiaging.	[113,114,115,116,117]

**Table 2 metabolites-15-00124-t002:** Clinical trials showing the effectiveness of some phytocompounds in neurodegenerative conditions.

Compound	Type of Study	Patients	Interventions	Outcomes	Ref.
Quercetin	Randomized, double-blind, placebo-controlled, and parallel group	Eighty healthy men and women.	500 mL bottled barley tea. The active group consumed a beverage containing 110 mg of quercetin. The placebo group consumed a beverage that did not contain quercetin.	Cognitive function, cognitive flexibility, executive function, and reaction time improved within the active group from baseline to week 40.	[220]
Huperzine A	Double-blind study	Fifty individuals with Alzheimer’s disease and 50 healthy individuals were used.	Patients received Huperzine-A medication via an oral dose of 0.2 mg twice daily for 8 weeks.	Alzheimer’s disease patients showed significant improvement in cognitive domains and task-switching skills.	[221,222]
Caffeine	Double-blind randomized placebo-controlled study	The study included 48 healthy, right-handed male participants (M = 26.27, SD = 3.47, range: 21–36).	Each participant received just one stimulant (200 mg of caffeine or placebo) in the form of a white capsule for oral ingestion and was tested on two different occasions at the same time of day in the early afternoon.	Positive effects were found on attention, cognitive control, and memory, including visual and auditory material.	[223,224,225]
Ginkgolides	Randomized, placebo-controlled, double-bind	60 subjects (50.8%) were in the ginkgolides group, and 58 subjects (49.2%) were in the placebo group.	The GBE group received 240 mg daily (80 mg, 3 times daily) for 42 months.	Protective effects against depression and dementia were found.	[226]
Curcumin	Randomized, placebo-controlled, double-bind	160 elderly patients were separated into either the curcumin group or the placebo group.	Subjects in the curcumin group received a single daily dose of 400 mg for 12 months.	Improvement in cognitive performance was observed.	[227,228]
Rosmarinic acid	Randomized, placebo-controlled, double-bind	A total of 23 patients were randomized to either the *M. officinalis *group (12 patients) or the placebo group (11 patients).	The patients received 500 mg of rosmarinic acid daily for 24 weeks.	The substance had a positive effect on agitation in patients with severe dementia.	[229,230]
Ferulic acid	Prospective, open-label study	20 patients, 4 men and 16 women aged 72–92 years.	Subjects were prescribed a dose of 3.0 g/day (morning and evening) of ferulic acid for 4 weeks.	The use of the compound showed more significant improvements in disinhibition, irritability/lability, aberrant behavior, hallucinations, and depression.	[231]
Resveratrol	Randomized, double-blind, and placebo-controlled	Forty-six (28 men, 18 women), 50–75 years, elderly healthy individuals.	Subjects were told to follow a daily intake of four capsules (200 mg of resveratrol) over 26 weeks. Subjects of the control group received placebo capsules.	Positive effect on memory functions (significant increase in word retention).	[232]

**Abbreviations:** GBE, *Ginkgo biloba* extract.

## Data Availability

This study did not create or analyze new data, and data sharing is not applicable to this article.

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
