# Peer review of "Polyphenols, Alkaloids, and Terpenoids Against Neurodegeneration: Evaluating the Neuroprotective Effects of Phytocompounds Through a Comprehensive Review of the Current Evidence"

_metabolites, 2025, doi:10.3390/metabo15020124_

Round 1
Reviewer 1 Report
Comments and Suggestions for Authors
Dear Authors.
Plants attract the attention of researchers as a source of various useful substances with biological activity. Plants are really rich in phytocomponents, including those that can help prevent neurodegenerative disorders. But the purpose of the review is not clear and its main idea is not disclosed. The authors' critical view of the problem under study is lacking. The article in its current form requires significant processing and is unsuitable for publication in Metabolites.
Major comments:
Introduction
1) The purpose of the review is not clear. "... to show the role of some phytochemicals in these conditions" - the principle of selection of phytochemicals? The criteria for selection are not clear.
2) What is the scientific novelty of the presented review? The list of sources includes 215 publications. Of these, a significant part were published in 2024 and 2025. What is the fundamental difference between the presented review and those already published?
2. Neurodegenerative conditions
3) The authors have formulated the existing problems. But they haven't identified any ways to overcome them? How can phytocomponents help solve the problems of neurodegenerative disorders?
4) L. 215 “This compound is used in the treatment of diseases such as hypertension, inflammatory disorders, and cancer [108,109]”. Such components are not used in the self-treatment of these diseases.
5) S. 3, check the numbering of subitems.
Figures 1-3
6) Figures are not perceived unambiguously. The principle of information on them is not clear.
7) There are no captions. There are explanations to the drawings, but there are no captions.
4. Conclusions and Future Research Perspectives
8) The conclusions should clearly correspond to the purpose of the review. Why talk about the potential of phytocomponents in the treatment of other diseases?
9) Bibliographic records of sources should be complete.
Author Response
RESPONSE TO REVIEWERS' COMMENTS
Manuscript number: metabolites-3417643― Metabolites (MDPI)
"Phytocompounds’ Effects on Neurodegeneration: A Comprehensive Review of the Current Evidence"
The authors of this document wish to express their deepest gratitude to the Editor-in-Chief and the Reviewer for their thorough and insightful evaluation of our manuscript. Their expert feedback has been invaluable in enhancing the quality of our work. We have carefully considered and diligently implemented each suggestion, significantly improving the manuscript. We have made substantial revisions to address the points raised. These noteworthy changes are marked mainly with YELLOW-highlighted text throughout the document for ease of reference. For corrections highlighted with a different color, there will be a note for the referee advertising. Additionally, we have prepared a detailed and comprehensive response to each comment and suggestion. This response is organized in a "point-by-point" format below, ensuring that every concern has been thoroughly addressed and explained. We sincerely appreciate the time and effort invested by the Editor-in-Chief and the Reviewer, and we believe their contributions have significantly strengthened the final version of our manuscript.
REVIEWER #1
General comment
Dear Authors. Plants attract the attention of researchers as a source of various useful substances with biological activity. Plants are really rich in phytocomponents, including those that can help prevent neurodegenerative disorders. But the purpose of the review is not clear and its main idea is not disclosed. The authors' critical view of the problem under study is lacking. The article in its current form requires significant processing and is unsuitable for publication in Metabolites.
General response
Dear Erudite Reviewer, thank you for taking the time to revise our manuscript and allowing us to improve based on your precious comments and suggestions. After addressing all your comments and suggestions regarding our manuscript text, we are confident that a significantly enhanced manuscript version has emerged. We are excited to resubmit the modified version for your perusal and reevaluation. Thank you for your brilliant insights, essential contributions, and feedback. You do have an eye for improvement. As a signal of our utmost respect for you, we want to provide you with a detailed and comprehensive point-by-point response to your comments below. Thank you once again for your time and patience in revising our article.
Comment #1
The purpose of the review is not clear. "... to show the role of some phytochemicals in these conditions" - the principle of selection of phytochemicals? The criteria for selection are not clear.
Response
Dear Erudite Reviewer, thank you for this comment. You are entirely correct, and we agree with you. We have included Lines 141-150 on Pages 3-4 of our introduction to overcome this limitation. These lines delve into the purpose of our review, making a more explicit statement that we paved the way for our literature exploration to fill a gap in the current literature, which is the inexistence of a comprehensive review covering the neuroprotective aspects of polyphenols, alkaloids, and terpenoids against neurodegeneration based on clinical studies.
Again, thank you for your brilliant suggestions. You have an eye for detail, and we are grateful for communicating with such an esteemed and critical reviewer.
Comment #2
What is the scientific novelty of the presented review? The list of sources includes 215 publications. Of these, a significant part were published in 2024 and 2025. What is the fundamental difference between the presented review and those already published?
Response
Dear Erudite Reviewer, thank you for this insightful suggestion. We agree with you, and to revise our manuscript accordingly, we have included Lines 126-150 on Pages 3-4 to insightfully discuss the novelty of our manuscript considering the already published research. We believe our manuscript has been enhanced after we addressed this suggestion. Thank you for your diligent patience and helpful guidance.
Comment #3
The authors have formulated the existing problems. But they haven't identified any ways to overcome them? How can phytocomponents help solve the problems of neurodegenerative disorders?
Response
Dear Erudite Reviewer, thank you for this comment. You are entirely correct, and we agree that we must have addressed the included studies broadly, discussing the effects profoundly and trying to familiarize the readers with the imposed future research directions based on each included study’s limitations. Therefore, we added Lines 760-819 on Pages 21-22 delving into the included studies’ discussion and limitations, translating findings into future research endeavors. We hope you can approve our manuscript now. We are confident that our text has been significantly improved since we addressed your comments and suggestions. We are thankful for the opportunity to have you assess our work and give us precious feedback. Thank you for everything!
Comment #4
- 215 “This compound is used in the treatment of diseases such as hypertension, inflammatory disorders, and cancer [108,109]”. Such components are not used in the self-treatment of these diseases.
Response
Dear Erudite Reviewer, thank you for this comment. You are correct; we agree that our sentence must have been more straightforward. Considering this, we reformulated the sentence to pursue the idea that luteolin, as a bioactive compound, may be beneficial as an adjuvant treatment against many human diseases, including hypertension, inflammatory disorders, and cancer. We believe this addition has significantly improved our manuscript. Please refer to Lines 247-250 on Page 7 for the additional content. We have included three additional references to our manuscript to reference our new sentence, especially right after the sentence. These are highlighted below for your kind review.
- Castellino, G.; Nikolic, D.; Magán-Fernández, A.; Malfa, G.A.; Chianetta, R.; Patti, A.M.; Amato, A.; Montalto, G.; Toth, P.P.; Banach, M.; et al. Altilix(®) Supplement Containing Chlorogenic Acid and Luteolin Improved Hepatic and Cardiometabolic Parameters in Subjects with Metabolic Syndrome: A 6 Month Randomized, Double-Blind, Placebo-Controlled Study. Nutrients 2019, 11, doi:10.3390/nu11112580.
- Gates, M.A.; Tworoger, S.S.; Hecht, J.L.; De Vivo, I.; Rosner, B.; Hankinson, S.E. A prospective study of dietary flavonoid intake and incidence of epithelial ovarian cancer. Int J Cancer 2007, 121, 2225-2232, doi:10.1002/ijc.22790.
- Hodgin, K.S.; Donovan, E.K.; Kekes-Szabo, S.; Lin, J.C.; Feick, J.; Massey, R.L.; Ness, T.J.; Younger, J.W. A Placebo-Controlled, Pseudo-Randomized, Crossover Trial of Botanical Agents for Gulf War Illness: Resveratrol (Polygonum cuspidatum), Luteolin, and Fisetin (Rhus succedanea). Int J Environ Res Public Health 2021, 18, doi:10.3390/ijerph18052483.
Thank you for your attention to detail and eye for improvement. It is a true honor to communicate with such an esteemed and critical reviewer.
Comment #5
- 3, check the numbering of subitems.
Response
Dear Erudite Reviewer, thank you for bringing this to our attention. You have an eye for detail and improvement. After carefully revising the order of the subsections in Section 3, we have implemented one correction. Anteriorly, Subsection “3.1.2. Quercetin” was numbered incorrectly. Therefore, after addressing your comment, we have diligently corrected the ordination of the Subsections. Please refer to Line 314 on Page 8 for the amendment. Thank you for your comments. Your help has been instrumental in reshaping our manuscript for the better.
Comment #6
Figures are not perceived unambiguously. The principle of information on them is not clear. There are no captions. There are explanations to the drawings, but there are no captions.
Response
Dear Erudite Reviewer, thank you for this comment. You are entirely correct, and we agree with you. Therefore, we have implemented modifications in Figures 1, 2, and 3 to address your concerns correctly. Please refer to Pages 10, 14, and 18 for the new and revised Figures 1, 2, and 3, respectively. In addition, we have also modified their legends, improving the understanding of the figures by directly addressing their principle of information, enhancing their captions, and implementing explanations in the drawings. Please refer to Lines 360-366 on Page 10, Lines 516-537 on Pages 14-15, and Lines 689-710 on Page 19 to review the newly added information within the Figures’ legends. You will review the Figures’ legends in their respective order. We hope you will approve the modified version of our document. Our manuscript has been pretty enhanced after we addressed your critical and vital suggestions. Your comments have been instrumental in reshaping our manuscript for the better; we eagerly anticipate your positive response.
Comment #7
The conclusions should clearly correspond to the purpose of the review. Why talk about the potential of phytocomponents in the treatment of other diseases?
Response
Dear Erudite Reviewer, you are entirely correct, and we thank you for this insightful suggestion. To improve our manuscript’s conclusions accordingly, we have deleted information regarding diseases other than those associated with neurodegeneration from those lines. Within Lines 821-855 on Page 23, you will find a conclusion section that only delves into the purpose of our review, which is the effects of phytocompounds against neurodegeneration; thank you for this critical suggestion. We believe our manuscript has been significantly improved after we addressed your concerns. We are proud of the revised version you helped us build!
Comment #8
Bibliographic records of sources should be complete.
Response
Dear Erudite Editor, you are correct, and we agree with your comment. To improve our manuscript accordingly, we have implemented modifications to the manuscript’s methodology. We have included Section 4, “Phytocompounds’ Effects on Neurodegeneration: A Comprehensive Overview Based on Clinical Trials,” (Lines 713-717 on Pages 19-20) with three distinct subsections. Please refer to Lines 718-743 on Page 20 for the complete bibliographic sources used in our manuscript written. In addition, you will find the Literature Search Report based on the above methodology within Lines 745-758 on Pages 20-21. Finally, in Lines 760-819 on Pages 21-22, you will find a complete discussion of the included randomized and non-randomized clinical trials in our review. We hope our manuscript meets your rigorous standards now, and we eagerly anticipate your positive response!
Again, thank you for your commitment to revising our manuscript. Your valuable input has undoubtedly enhanced its quality, and we are proud to present this revised version for your review and approval!
I, the corresponding author of the manuscript "Phytocompounds’ Effects on Neurodegeneration: A Comprehensive Review of the Current Evidence" under the assigned ID metabolites-3417643, on behalf of my coauthors, once again extend my heartfelt gratitude to the knowledgeable Editor-in-Chief and reviewers for their time and expertise in revising our manuscript. After we addressed their constructive and refined feedback and suggestions, a significantly improved manuscript version emerged. Undoubtedly, their insightful suggestions and feedback have significantly enhanced the quality of our manuscript. We respectfully are at the disposal of the Editor-in-Chief and the Reviewer to address any additional suggestions regarding our publication. Suppose you are satisfied with our newly refined and significantly improved version. In that case, we are eager to anticipate the acceptance of our article for publication in this most critical journal, Metabolites. Thank you once again for your time and expertise.
Reviewer 2 Report
Comments and Suggestions for Authors
This review was written about phytocompounds’ effects on neurodegeneration: a comprehensive review of the current evidence. Recently, there have been reports about phytochemicals with neurodegeneration effects. However, there is not much data collected from clinical studies. Please add more data about the clinical study to show more evidence from phytocompounds, it would make your review more interesting. Moreover, these points below should be asked to the authors
1. Please provide the reasons for the selection of the phytocompounds mentioned in the manuscript. Because other phytocompounds have also been reported to exhibit neurodegeneration effects.
2. Line 36: Terpenoids are not alkaloids and should be removed from parentheses.
3. Line 36-37: Since flavonoids are also polyphenolic substances, why are flavonoids not included in this group? If luteolin and quercetin are classified as flavonoids, resveratrol and pterostilbene should be classified as stilbene as well.
4. Line 218: Please check the number in the chemical formula because it should be subscript.
5. Line 245: Please check the spelling of augmnt.
6. Line 319, 491: Please add more sentences after the first sentence of the paragraph.
7. Figure 1 Please show the mechanism underlying neurodegeneration effects of each phytocompounds. The details are like those shown in Figure 2 and Figure 3.
8. Figure 1: The diagram has a + sign showing that Luteolin is used together with Quercetin to achieve the antioxidant effect.
9. Page 12: Please check that curcumin is a terpenoid group, is it correct or not?
10. Please discuss the efficacy and safety of the phytocompounds from the clinical study that shown in Table 2.
11. Please check the spelling of phytocompounds that should be the same spelling without space (phyto_compounds).
Author Response
RESPONSE TO REVIEWERS' COMMENTS
Manuscript number: metabolites-3417643― Metabolites (MDPI)
"Phytocompounds’ Effects on Neurodegeneration: A Comprehensive Review of the Current Evidence"
The authors of this document wish to express their deepest gratitude to the Editor-in-Chief and the Reviewer for their thorough and insightful evaluation of our manuscript. Their expert feedback has been invaluable in enhancing the quality of our work. We have carefully considered and diligently implemented each suggestion, significantly improving the manuscript. We have made substantial revisions to address the points raised. These noteworthy changes are marked mainly with YELLOW-highlighted text throughout the document for ease of reference. For corrections highlighted with a different color, there will be a note for the referee advertising. Additionally, we have prepared a detailed and comprehensive response to each comment and suggestion. This response is organized in a "point-by-point" format below, ensuring that every concern has been thoroughly addressed and explained. We sincerely appreciate the time and effort invested by the Editor-in-Chief and the Reviewer, and we believe their contributions have significantly strengthened the final version of our manuscript.
REVIEWER #2
General comment
This review was written about phytocompounds’ effects on neurodegeneration: a comprehensive review of the current evidence. Recently, there have been reports about phytochemicals with neurodegeneration effects. However, there is not much data collected from clinical studies. Please add more data about the clinical study to show more evidence from phytocompounds, it would make your review more interesting. Moreover, these points below should be asked to the authors
General response
Dear Erudite Reviewer, thank you for taking the time to revise our manuscript and allowing us to improve based on your precious comments and suggestions. After addressing all your comments and suggestions regarding our manuscript text, we are confident that a significantly enhanced manuscript version has emerged. We are excited to resubmit the modified version for your perusal and reevaluation. Thank you for your brilliant insights, essential contributions, and feedback. You do have an eye for improvement. As a signal of our utmost respect for you, we want to provide you with a detailed and comprehensive point-by-point response to your comments below. Thank you once again for your time and patience in revising our article.
Comment #1
Please provide the reasons for the selection of the phytocompounds mentioned in the manuscript. Because other phytocompounds have also been reported to exhibit neurodegeneration effects.
Response
Dear Erudite Reviewer, thank you for this insightful comment and suggestion. You are entirely correct, and we agree that adding comprehensive details regarding our literature review would undoubtedly increase the rigor and transparency of our findings, highlighting the significance of our work. Therefore, based on your precious input, we included Lines 141-150 on Pages 3-4 of the revised document to correct our manuscript. Thank you for your contributions. Your guidance has been instrumental in reshaping our manuscript for the better. We eagerly anticipate your positive response and the acceptance of the publication of our manuscript in this excellent journal. Thank you for everything!
Comment #2
Line 36: Terpenoids are not alkaloids and should be removed from parentheses.
Response
Dear Erudite Reviewer, thank you for this comment. You have an eye for improvement and attention to detail. We agree with you. To correct the manuscript accordingly, we rephrased the sentence. Please refer to Lines 90-92 on Page 2 for the corrected sentence in the revised manuscript document. Thank you for your patience and guidance!
Comment #3
Line 36-37: Since flavonoids are also polyphenolic substances, why are flavonoids not included in this group? If luteolin and quercetin are classified as flavonoids, resveratrol and pterostilbene should be classified as stilbene as well.
Response
Dear Erudite Reviewer, thank you for this comment. We appreciate your precious input, which enhanced our manuscript’s quality and readability. We agree with you. To correct the manuscript accordingly, we rephrased the abstract. Please refer to Lines 24-52 on Pages 1-2 for the corrected sentence in the revised manuscript document. Thank you for your patience and guidance! You have an eye for improvement, and we are proud to present this revised version of our manuscript for your kind review.
Comment #4
Line 218: Please check the number in the chemical formula because it should be subscript.
Response
Dear Erudite Reviewer, thank you for this comment. You have an eye for detail and improvement. We have corrected the subscript in the chemical formula in Line 250 on Page 7 of the revised manuscript document. Again, thank you for your attention. You truly enhanced our manuscript’s quality and readability.
Comment #5
Line 245: Please check the spelling of augmnt.
Response
Dear Erudite Reviewer, thank you for this comment. You are entirely correct, and we agree with you. To correct our manuscript accordingly, we modified the sentence in Lines 278-279 on Page 7 of the revised manuscript document. Your comment substantially enhanced our manuscript’s quality and readability. Thank you for your attention to detail and eye for improvement.
Comment #6
Line 319, 491: Please add more sentences after the first sentence of the paragraph.
Response
Dear Erudite Reviewer, thank you for this comment. You are entirely correct, and we agree that adding more sentences right after introducing the figures in our manuscript’s main document may be necessary. To improve our manuscript accordingly, we diligently increased the above-mentioned paragraphs with brief highlights from the two figures. Please refer to Lines 353-358 on Page 9 for the first augmentation and Lines 508-514 on Page 13 for the second augmentation. We believe our manuscript has been significantly improved after we addressed these issues, and your input has been instrumental in reshaping our manuscript for the better. Thank you for everything!
Comment #7
Figure 1 Please show the mechanism underlying neurodegeneration effects of each phytocompounds. The details are like those shown in Figure 2 and Figure 3.
Response
Dear Erudite Reviewer, thank you for this brilliant suggestion and precious input. You are entirely correct that adding the underlying mechanisms behind the phytochemicals in Figure 1 would undoubtedly enhance our manuscript’s quality and readability. To correct our manuscript accordingly, we modified Figure 1 and included the revised version in the document. Please refer to the new Figure 1 on Page 10. The underlying mechanisms are in the orange and green ballots. This is to facilitate your review of our revised manuscript version. We eagerly anticipate your positive response and the approval of our manuscript for publication in this critical journal. Again, thank you for your collaboration and guidance!
Comment #8
Figure 1: The diagram has a + sign showing that Luteolin is used together with Quercetin to achieve the antioxidant effect.
Response
Dear Erudite Reviewer, thank you for this comment. We must say that the “+” sign did not mean anything in Figure 1. It does not represent additional or synergistic effects between the studied compounds. Therefore, to improve our manuscript accordingly, we deleted the “+” sign from Figure 1 in the revised manuscript document. Please refer to Page 10 for the new and revised Figure 1. Thank you for your precious input. You have an eye for improvement, and it is a true honor for us to communicate with such an esteemed reviewer.
Comment #9
Page 12: Please check that curcumin is a terpenoid group, is it correct or not?
Response
Dear Erudite Reviewer, thank you for this comment of utmost importance. Curcumin is a polyphenol, and we have now positioned the paragraph regarding this phytochemical in the right section. Importantly, the paragraphs about curcumin are highlighted in BLUE-highlighted text in Lines 661-681 on Pages 17-18. This is to facilitate your kind review and demonstrate that no modifications have been made this time. Thank you for your attention to detail and eye for improvement. Your contributions have significantly improved the quality and presentation of our manuscript. Thank you for everything!
Comment #10
Please discuss the efficacy and safety of the phytocompounds from the clinical study that shown in Table 2.
Response
Dear Erudite Reviewer, thank you for this comment. You are entirely correct, and we agree that adding more detail about the included studies would undoubtedly enhance our manuscript’s quality and readability. Therefore, in Lines 760-819 on Pages 21-22, you will find a complete discussion of the included randomized and non-randomized clinical trials in our review. In these lines, we discuss every study included and gather the evidence to familiarize the readers with the concepts we’ve discussed even more. Thank you for your critical appraisal of our manuscript. It is a true honor to have our manuscript assessed by a crucial reviewer like you. Thank you for everything!
Comment #11
Please check the spelling of phytocompounds that should be the same spelling without space (phyto_compounds).
Response
Dear Erudite Reviewer, thank you for this comment. You have attentive eyes! We revised the entire manuscript carefully to correct errors like the ones mentioned above. “Phyto compounds” has been corrected into “phytocompounds” across the manuscript’s sections. Please refer to Line 34 on Page 1 for the first correction, Line 36 on Page 1 for the second correction, Lines 40-41 on Page 1 for the third correction, Line 116 on Page 3 for the fourth correction, and Line 344 on Page 9 for the fifth correction. Again, thank you for your brilliant input. Your comments and suggestions have been instrumental in reshaping our manuscript for the better. We eagerly anticipate your positive response and the approval of our manuscript for publication in this essential journal.
I, the corresponding author of the manuscript "Phytocompounds’ Effects on Neurodegeneration: A Comprehensive Review of the Current Evidence" under the assigned ID metabolites-3417643, on behalf of my coauthors, once again extend my heartfelt gratitude to the knowledgeable Editor-in-Chief and reviewers for their time and expertise in revising our manuscript. After we addressed their constructive and refined feedback and suggestions, a significantly improved manuscript version emerged. Undoubtedly, their insightful suggestions and feedback have significantly enhanced the quality of our manuscript. We respectfully are at the disposal of the Editor-in-Chief and the Reviewer to address any additional suggestions regarding our publication. Suppose you are satisfied with our newly refined and significantly improved version. In that case, we are eager to anticipate the acceptance of our article for publication in this most critical journal, Metabolites. Thank you once again for your time and expertise.
Reviewer 3 Report
Comments and Suggestions for Authors
Title: Phytocompounds’ Effects on Neurodegeneration: A Comprehensive Review of the Current Evidence
Overview; Reviewer’s comments
This review article focused on the Current Evidence of the Phytocompounds’ Effects on Neurodegeneration. The compounds reviewed included flavonoids (luteolin and quercetin), Alkaloids (berberine and Huperzine A, caffeine, and terpenoids), phenolic acids (caffeic acid, ferulic acid, and pterostilbene), and polyphenols (resveratrol, and pterostilbene). The study excellently summarized that compounds that are at clinical level. In summary, the compounds were related to significant improvements in cognition, memory, disinhibition, irritability/lability, aberrant behavior, hallucinations, and depression. There is absolutely no doubt that this comprehensive review will add knowledge of knowledge, subject to the recommended corrections.
Abstract
Comment 1: The abstract focused too much on literature data to introduce and justify the need for this type of study. However, very little attention was given to methodology used in the review, synopsis of perhaps the most promising phytocompounds against neurogenerative disorders. The abstract could also give an indication, ff there are phytocompounds or trends of certain class(es) of compounds that were investigated up to clinical level. In this current form, it is not clear which classes of phytochemicals were interesting and how far the evidence has gone i.e. in vitro evidence or is there any clinical evidence of the potential use of phytochemicals to treat neurogenerative disorders.
Introduction
Comment 1: On Line 79, the authors must write ROS in full since it has been mentioned for the first time.
Methodology
Comment 2: Its not clear from the study how the reviewed phytocompounds were selected for this study since there is no methodology section. Secondly, it will be interesting to indicate that time frames that the review covers. Therefore, the authors must include a detailed methodology section. Specify the period of study (duration in years), the sources of literature (e.g., internet-based or printed materials), and the criteria used for selecting or excluding phytochemicals for the review.
Discussions
Comment 5: The detailed discussions of each compound are very compelling and excellently written. However, a summarized discussion at the end to extrapolates the hierarchy of the importance of the discussed compounds in relation to the progress made in their potential to treat neurogenerative disorder will give readers an insight into which compounds are showing great potential based on the evidence gathered from literature. The summarized discussion will then be a very good build up to the excellently written conclusions and future perspectives by the authors.
Comment 6: Table 2 appears to not have been referenced in the texts. Please verify.
Comment 7: Based on Table 2, it is evident that the study primarily focused on phyto-compounds that are already or have passed the clinical trial. Perhaps the authors should also consider including this point in inclusion and exclusion criteria for the study in the methodology section.
Author Response
RESPONSE TO REVIEWERS' COMMENTS
Manuscript number: metabolites-3417643― Metabolites (MDPI)
"Phytocompounds’ Effects on Neurodegeneration: A Comprehensive Review of the Current Evidence"
The authors of this document wish to express their deepest gratitude to the Editor-in-Chief and the Reviewer for their thorough and insightful evaluation of our manuscript. Their expert feedback has been invaluable in enhancing the quality of our work. We have carefully considered and diligently implemented each suggestion, significantly improving the manuscript. We have made substantial revisions to address the points raised. These noteworthy changes are marked mainly with YELLOW-highlighted text throughout the document for ease of reference. For corrections highlighted with a different color, there will be a note for the referee advertising. Additionally, we have prepared a detailed and comprehensive response to each comment and suggestion. This response is organized in a "point-by-point" format below, ensuring that every concern has been thoroughly addressed and explained. We sincerely appreciate the time and effort invested by the Editor-in-Chief and the Reviewer, and we believe their contributions have significantly strengthened the final version of our manuscript.
REVIEWER #3
General comment
This review article focused on the Current Evidence of the Phytocompounds’ Effects on Neurodegeneration. The compounds reviewed included flavonoids (luteolin and quercetin), Alkaloids (berberine and Huperzine A, caffeine, and terpenoids), phenolic acids (caffeic acid, ferulic acid, and pterostilbene), and polyphenols (resveratrol, and pterostilbene). The study excellently summarized that compounds that are at clinical level. In summary, the compounds were related to significant improvements in cognition, memory, disinhibition, irritability/lability, aberrant behavior, hallucinations, and depression. There is absolutely no doubt that this comprehensive review will add knowledge of knowledge, subject to the recommended corrections.
General response
Dear Erudite Reviewer, thank you for taking the time to revise our manuscript and allowing us to improve based on your precious comments and suggestions. After addressing all your comments and suggestions regarding our manuscript text, we are confident that a significantly enhanced manuscript version has emerged. We are excited to resubmit the modified version for your perusal and reevaluation. Thank you for your brilliant insights, essential contributions, and feedback. You do have an eye for improvement. As a signal of our utmost respect for you, we want to provide you with a detailed and comprehensive point-by-point response to your comments below. Thank you once again for your time and patience in revising our article.
Comment #1
The abstract focused too much on literature data to introduce and justify the need for this type of study. However, very little attention was given to methodology used in the review, synopsis of perhaps the most promising phytocompounds against neurogenerative disorders. The abstract could also give an indication, ff there are phytocompounds or trends of certain class(es) of compounds that were investigated up to clinical level. In this current form, it is not clear which classes of phytochemicals were interesting and how far the evidence has gone i.e. in vitro evidence or is there any clinical evidence of the potential use of phytochemicals to treat neurogenerative disorders.
Response
Dear Erudite Reviewer, thank you for this insightful suggestion and precious input. You are entirely correct, and we agree that adding more information regarding our article to the manuscript would undoubtedly enhance our document’s quality and readability. To improve our manuscript, we diligently modified the abstract in Lines 24-51 on Pages 1-2 of the revised manuscript’s document. We are thankful for the opportunity to communicate with such an esteemed reviewer, and we eagerly anticipate a positive response from you and the approval of our manuscript for publication in this excellent journal.
Comment #2
On Line 79, the authors must write ROS in full since it has been mentioned for the first time.
Response
Dear Erudite Reviewer, thank you for this insightful suggestion. You have an eye for detail and improvement. To revise our manuscript accordingly, we have modified the sentence in Lines 90-92 on Page 2. Our manuscript has been significantly improved after we addressed your suggestions. Thank you for everything! We are grateful to communicate with such an esteemed reviewer.
Comment #3
Its not clear from the study how the reviewed phytocompounds were selected for this study since there is no methodology section. Secondly, it will be interesting to indicate that time frames that the review covers. Therefore, the authors must include a detailed methodology section. Specify the period of study (duration in years), the sources of literature (e.g., internet-based or printed materials), and the criteria used for selecting or excluding phytochemicals for the review.
Response
Dear Erudite Editor, you are correct, and we agree with your comment. To improve our manuscript accordingly, we have implemented modifications to the manuscript’s methodology. We have included Section 4, “Phytocompounds’ Effects on Neurodegeneration: A Comprehensive Overview Based on Clinical Trials,” (Lines 713-717 on Pages 19-20) with three distinct subsections. Please refer to Lines 718-743 on Page 20 for the complete bibliographic sources used in our manuscript written and the manuscript’s methodology. In addition, you will find the Literature Search Report based on the above methods within Lines 745-758 on Pages 20-21. Finally, in Lines 760-819 on Pages 21-22, you will find a complete discussion of the included randomized and non-randomized clinical trials in our review. We hope our manuscript meets your rigorous standards now, and we eagerly anticipate your positive response!
Again, thank you for your commitment to revising our manuscript. Your valuable input has undoubtedly enhanced its quality, and we are proud to present this revised version for your review and approval!
Comment #4
The detailed discussions of each compound are very compelling and excellently written. However, a summarized discussion at the end to extrapolates the hierarchy of the importance of the discussed compounds in relation to the progress made in their potential to treat neurogenerative disorder will give readers an insight into which compounds are showing great potential based on the evidence gathered from literature. The summarized discussion will then be a very good build up to the excellently written conclusions and future perspectives by the authors.
Response
Dear Erudite Reviewer, thank you for this comment. You are entirely correct, and we agree that adding more detail about the included studies would undoubtedly enhance our manuscript’s quality and readability. Therefore, in Lines 760-819 on Pages 21-22, you will find a complete discussion of the included randomized and non-randomized clinical trials in our review. In these lines, we discuss every study included and gather the evidence to familiarize the readers with the concepts we’ve discussed even more. These lines are right before the conclusions, which you highlighted. Thank you for your critical appraisal of our manuscript. It is a true honor to have our manuscript assessed by a crucial reviewer like you. Thank you for everything!
Comment #5
Table 2 appears to not have been referenced in the texts. Please verify.
Response
Dear Erudite Reviewer, thank you for this suggestion. We verified it and are happy to share that Table 2 was first cited in Lines 602-685 on Page 18 of our document. Since no modification has been made to this comment, we highlighted the sentence in BLUE-highlighted text. Thank you for your attention to detail and eye for improvement! We eagerly anticipate a positive response from you to our revised manuscript document.
Comment #6
Based on Table 2, it is evident that the study primarily focused on phyto-compounds that are already or have passed the clinical trial. Perhaps the authors should also consider including this point in inclusion and exclusion criteria for the study in the methodology section.
Response
Dear Erudite Editor, we agree with your comment. You are entirely correct at this point. To improve our manuscript accordingly, we have included Section 4, “Phytocompounds’ Effects on Neurodegeneration: A Comprehensive Overview Based on Clinical Trials,” (Lines 713-717 on Pages 19-20) with three distinct subsections. Please refer to Lines 718-744 on Page 20 for the complete manuscript’s methodology. Communicating with such a critical and esteemed reviewer is a true honor. Thank you for everything!
I, the corresponding author of the manuscript "Phytocompounds’ Effects on Neurodegeneration: A Comprehensive Review of the Current Evidence" under the assigned ID metabolites-3417643, on behalf of my coauthors, once again extend my heartfelt gratitude to the knowledgeable Editor-in-Chief and reviewers for their time and expertise in revising our manuscript. After we addressed their constructive and refined feedback and suggestions, a significantly improved manuscript version emerged. Undoubtedly, their insightful suggestions and feedback have significantly enhanced the quality of our manuscript. We respectfully are at the disposal of the Editor-in-Chief and the Reviewer to address any additional suggestions regarding our publication. Suppose you are satisfied with our newly refined and significantly improved version. In that case, we are eager to anticipate the acceptance of our article for publication in this most critical journal, Metabolites. Thank you once again for your time and expertise.
Round 2
Reviewer 1 Report
Comments and Suggestions for Authors
Dear Authors
My comments have been taken into account
Author Response
RESPONSE TO REVIEWERS' COMMENTS
Manuscript number: metabolites-3417643― Metabolites (MDPI)
"Polyphenols, Alkaloids, and Terpenoids against Neurodegeneration: Evaluating the Neuroprotective Effects of Phytocompounds through A Comprehensive Review of the Current Evidence "
REVIEWER #1
General comment
Dear Authors, my comments have been taken into account.
General response
Dear Esteemed Reviewer, thank you for taking the time to review our manuscript and for providing invaluable comments and suggestions. We have carefully addressed all your feedback, and we are grateful that the revised version of our manuscript was significantly improved. We are excited about the acceptance of our manuscript for publication in this critical journal. Your insights and contributions were instrumental in this process, and we truly appreciate your attention to detail. Thank you once again for your time and patience in reviewing our article.
Best regards,
The Authors.
Reviewer 2 Report
Comments and Suggestions for Authors
The revised manuscript about “Polyphenols, alkaloids, and terpenoids against neurodegeneration: Evaluating the neuroprotective effects of phytocompounds through a comprehensive review of the current evidence” has been improved but there are some points below that should be asked the authors.
Please check the group of compounds. In the abstract, resveratrol is classified as a phenolic acid group, but on page 16 it is classified as a polyphenol group. Additionally, in the conclusion part, phenolic acid and polyphenol are classified as separate groups, as in pages 15-17. However, in the abstract, it says polyphenol contains phenolic acid as a subgroup.
Figure 1:
1. Please specify what the name of the plant near the Luteolin is.
2. Berberine and Huperzine A exert anti-inflammatory, antioxidant, anti-apoptotic, neuro-stimulant, and pain control, but it does not correlate in table 1. There are no neuro-stimulant effects shown in table 1.
Figure 2:
Please check the plants photo below gingolide and curcumin because they are swapped.
Figure 3:
Why does Resveratrol have no inhibition of NF-kB activity? Please check in this reference “Feng, Y., Cui, Y., Gao, J., Li, M., Li, R., Jiang, X., Tian, Y., Wang, K., Cui, C., & Cui, J. (2016). Resveratrol attenuates neuronal autophagy and inflammatory injury by inhibiting the TLR4/NF-κB signaling pathway in experimental traumatic brain injury. International Journal of Molecular Medicine, 37(4), 921–930. https://doi.org/10.3892/ijmm.2016.2495.”
Author Response
RESPONSE TO REVIEWERS' COMMENTS
Manuscript number: metabolites-3417643― Metabolites (MDPI)
"Polyphenols, Alkaloids, and Terpenoids against Neurodegeneration: Evaluating the Neuroprotective Effects of Phytocompounds through A Comprehensive Review of the Current Evidence "
The authors of this document wish to express their deepest gratitude to the Editor-in-Chief and the Reviewer for their thorough and insightful evaluation of our manuscript. Their expert feedback has been invaluable in enhancing the quality of our work. We have carefully considered and diligently implemented each suggestion, significantly improving the manuscript. We have made substantial revisions to address the points raised. These noteworthy changes are marked mainly with YELLOW-highlighted text throughout the document for ease of reference. For corrections highlighted with a different color, there will be a note for the referee advertising. Additionally, we have prepared a detailed and comprehensive response to each comment and suggestion. This response is organized in a "point-by-point" format below, ensuring that every concern has been thoroughly addressed and explained. We sincerely appreciate the time and effort invested by the Editor-in-Chief and the Reviewer, and we believe their contributions have significantly strengthened the final version of our manuscript.
REVIEWER #2
General comment
The revised manuscript about “Polyphenols, alkaloids, and terpenoids against neurodegeneration: Evaluating the neuroprotective effects of phytocompounds through a comprehensive review of the current evidence” has been improved but there are some points below that should be asked the authors.
General response
Dear Reviewer, thank you for taking the time to review our manuscript and for your invaluable comments and suggestions. We have carefully addressed all your feedback, and we believe that it has greatly improved the quality of our manuscript. We are excited to resubmit the revised version for your consideration and evaluation. Your insightful contributions and suggestions have been instrumental in this process, and we truly appreciate your keen eye for improvement. As a testament to our respect for your work, we have provided a detailed, point-by-point response to your comments below. Thank you once again for your time and patience in revising our article.
Comment #1
Please check the group of compounds. In the abstract, resveratrol is classified as a phenolic acid group, but on page 16 it is classified as a polyphenol group. Additionally, in the conclusion part, phenolic acid and polyphenol are classified as separate groups, as in pages 15-17. However, in the abstract, it says polyphenol contains phenolic acid as a subgroup.
Response
Dear Erudite Reviewer, thank you for this essential suggestion. We agree with you as you are entirely correct, and we are thankful for the opportunity to correct our manuscript based on your precious input. Therefore, we corrected the abstract in Lines 42-45 on Page 1. Now, the grammar of the sentence was corrected to include resveratrol not as a phenolic acid, which this compound is not, but as a polyphenol in general, which this compound truly is. Additionally, in the discussion part and conclusion section, polyphenols have been described as a unique group, encompassing all polyphenols described in our manuscript together. Please see the modifications within the subsection and section headings highlighted in Line 414 on Page 11, Line 415 on Page 11, Line 445 on Page 11, Line 469 on Page 12, Line 489 on Page 12, Line 507 on Page 13, Line 536 on Page 13, Line 606 on Page 15, and Line 660 on Page 16 of the revised manuscript’s document for the corrections within the discussion. For corrections to the conclusions, please refer to Lines 820-825 on Page 23. The conclusions’ first sentences have been corrected to include the name of the phytocompounds’ classes as it is in our manuscript’s title, “polyphenols, alkaloids, and terpenoids.”
Again, thank you for your expertise and time spent assessing our manuscript accordingly. We are thankful for the opportunity to communicate with such an esteemed reviewer. We hope our manuscript meets your standards for publication in this critical and essential journal.
Comment #2
Figure 1:
- Please specify what the name of the plant near the Luteolin is.
- Berberine and Huperzine A exert anti-inflammatory, antioxidant, anti-apoptotic, neuro-stimulant, and pain control, but it does not correlate in table 1. There are no neuro-stimulant effects shown in table 1.
Response
Dear Erudite Reviewer, thank you for this insightful suggestion and critical comment. You are entirely correct, and we agree that implementing corrections to Figure 2 (previous Figure 1) would undoubtedly enhance our figure’s quality and readability. Therefore, we included the name of the luteolin plant in our figure. Additionally, we removed “neuro-stimulant” from berberine and huperzine’s effects list. Please refer to Page 16 for the corrected Figure 2 (previous Figure 1). Thank you for your attention to detail and eye for improvement. Your contributions have been essential in reshaping our manuscript for the better. Thank you for everything!
Comment #3
Figure 2:
Please check the plants photo below gingolide and curcumin because they are swapped.
Response
Dear Erudite Reviewer, thank you for your attention to detail. You are entirely correct, and we agree with you that ginkgolides and Curcuma plants were swapped in Figure 1 (previous Figure 2). Therefore, we modified the figure to correct the position of the photos. Please refer to Page 10 for the corrected Figure 1 (previous Figure 2). We are thankful for the opportunity to have you assess our work, and we are eager to anticipate your positive response to the revised manuscript’s version.
Comment #4
Figure 3:
Why does Resveratrol have no inhibition of NF-kB activity? Please check in this reference “Feng, Y., Cui, Y., Gao, J., Li, M., Li, R., Jiang, X., Tian, Y., Wang, K., Cui, C., & Cui, J. (2016). Resveratrol attenuates neuronal autophagy and inflammatory injury by inhibiting the TLR4/NF-κB signaling pathway in experimental traumatic brain injury. International Journal of Molecular Medicine, 37(4), 921–930. https://doi.org/10.3892/ijmm.2016.2495.”
Response
Dear Erudite Reviewer, thank you for this insightful comment. You are correct, and we thank you for this suggestion. Therefore, we implemented modifications in Figure 3 on Page 18 to address your concerns correctly. Additionally, we modified the figure’s legend in Lines 702-705 on Page 19 to reflect the newly added content. Thank you for your comments and suggestions. Thank you for everything!
I, the corresponding author of the manuscript "Polyphenols, Alkaloids, and Terpenoids against Neurodegeneration: Evaluating the Neuroprotective Effects of Phytocompounds through A Comprehensive Review of the Current Evidence" under the assigned ID metabolites-3417643, on behalf of my coauthors, once again extend my heartfelt gratitude to the knowledgeable Editor-in-Chief and reviewers for their time and expertise in revising our manuscript. After we addressed their constructive and refined feedback and suggestions, a significantly improved manuscript version emerged. Undoubtedly, their insightful suggestions and feedback have significantly enhanced the quality of our manuscript. We respectfully are at the disposal of the Editor-in-Chief and the Reviewer to address any additional suggestions regarding our publication. Suppose you are satisfied with our newly refined and significantly improved version. In that case, we are eager to anticipate the acceptance of our article for publication in this most critical journal, Metabolites. Thank you once again for your time and expertise.
Reviewer 3 Report
Comments and Suggestions for Authors
Congratulations to the authors. They addressed all the comments very well.
Author Response
RESPONSE TO REVIEWERS' COMMENTS
Manuscript number: metabolites-3417643― Metabolites (MDPI)
"Polyphenols, Alkaloids, and Terpenoids against Neurodegeneration: Evaluating the Neuroprotective Effects of Phytocompounds through A Comprehensive Review of the Current Evidence "
REVIEWER #3
General comment
Congratulations to the authors. They addressed all the comments very well.
General response
Dear Esteemed Reviewer, thank you for taking the time to review our manuscript and for providing invaluable comments and suggestions. We have carefully addressed all your feedback, and we are grateful that the revised version of our manuscript was significantly improved. We are excited about the acceptance of our manuscript for publication in this critical journal. Your insights and contributions were instrumental in this process, and we truly appreciate your attention to detail. Thank you once again for your time and patience in reviewing our article.
Best regards,
The Authors.